# Bayesian Learning-driven Prototypical Contrastive Loss for Class-Incremental Learning

**Nisha L. Raichur**                                     *nisha.lakshmana.raichur@iis.fraunhofer.de*
*Fraunhofer Institute for Integrated Circuits IIS, Nürnberg, Germany*

**Lucas Heublein**                                               *lucas.heublein@iis.fraunhofer.de*
*Fraunhofer Institute for Integrated Circuits IIS, Nürnberg, Germany*

**Tobias Feigl**                                                     *tobias.feigl@iis.fraunhofer.de*
*Fraunhofer Institute for Integrated Circuits IIS, Nürnberg, Germany*

**Alexander Rügamer**                                       *alexander.ruegamer@iis.fraunhofer.de*
*Fraunhofer Institute for Integrated Circuits IIS, Nürnberg, Germany*

**Christopher Mutschler**                             *christopher.mutschler@iis.fraunhofer.de*
*Fraunhofer Institute for Integrated Circuits IIS, Nürnberg, Germany*

**Felix Ott**                                                           *felix.ott@iis.fraunhofer.de*
*Fraunhofer Institute for Integrated Circuits IIS, Nürnberg, Germany*

**Reviewed on OpenReview:** *https://openreview.net/forum?id=dNWaTuKV9M*

## Abstract

The primary objective of methods in continual learning is to learn tasks in a sequential manner over time (sometimes from a stream of data), while mitigating the detrimental phenomenon of catastrophic forgetting. This paper proposes a method to learn an effective representation between previous and newly encountered class prototypes. We propose a prototypical network with a Bayesian learning-driven contrastive loss (BLCL), tailored specifically for class-incremental learning scenarios. We introduce a contrastive loss that incorporates novel classes into the latent representation by reducing intra-class and increasing inter-class distance. Our approach dynamically adapts the balance between the cross-entropy and contrastive loss functions with a Bayesian learning technique. Experimental results conducted on the CIFAR-10, CIFAR-100, and ImageNet100 datasets for image classification and images of a GNSS-based dataset for interference classification validate the efficacy of our method, showcasing its superiority over existing state-of-the-art approaches. **Git:** https://gitlab.cc-asp.fraunhofer.de/darcy_gnss/gnss_class_incremental_learning

## 1 Introduction

Numerous practical vision applications require the learning of new visual capabilities while maintaining high performance on existing ones. Examples include construction safety, employing reinforcement learning methodologies (Kirkpatrick et al., 2017), or adapting to novel types of interference (as image classification task) in global navigation satellite system (GNSS) operations (Ott et al., 2024; Brieger et al., 2022; Raichur et al., 2022; van der Merwe et al., 2023; Heublein et al., 2024b; Gaikwad et al., 2024; Manjunath et al., 2025). *Continual Learning* describes the setting of sequentially learning novel tasks while avoiding *catastrophic forgetting*, i.e., forgetting how to perform well on previously seen tasks (Kirkpatrick et al., 2017), which commonly occurs when models are trained successively on multiple tasks without revisiting earlier tasks. Recent methodologies try to circumvent catastrophic forgetting, for instance by utilization of feature extraction and fine-tuning adaptation techniques (Li & Hoiem, 2018), by leveraging off-policy learning

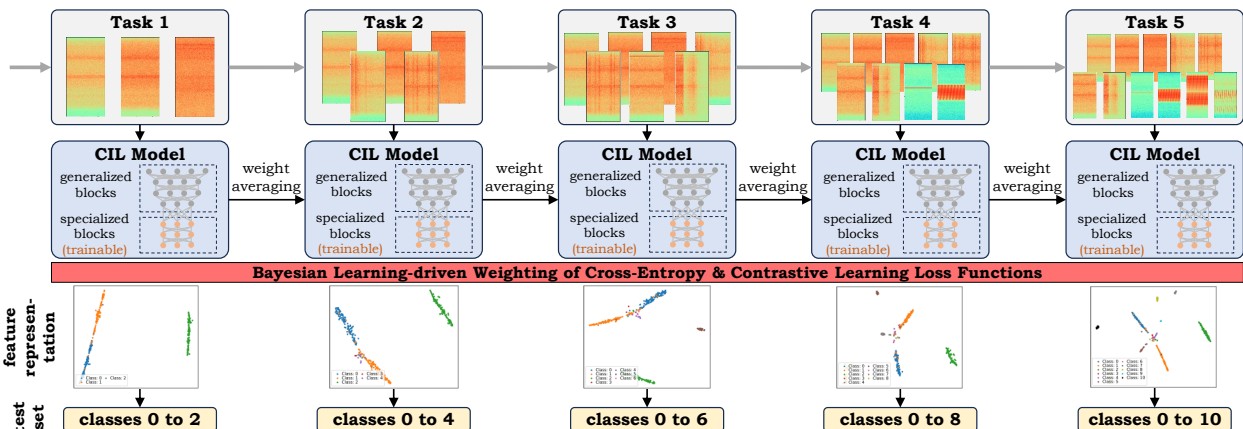

Figure 1: Overview of our BLCL approach, which employs Bayesian weighting to balance cross-entropy and contrastive learning loss functions, facilitating efficient representations during updating specialized blocks.

and behavioral cloning from replay to enhance stability (Rolnick et al., 2019), or by selectively decelerating learning on weights important for specific tasks (Kirkpatrick et al., 2017).

Previous work (Zhou et al., 2023b) investigated the effect of model size of the specialized components, i.e., shallow and deep layers, and concluded that deeper layers obtain diverse feature representations and is crucial for learning novel tasks. Specifically, the learned data representations between preceding and novel tasks hold considerable importance (Rebuffi et al., 2017). This serves as a motivation for our focus on refining the architecture of specialized blocks and enhancing dynamic representations for continual learning. Our proposed method, BLCL, enhances task adaptability by dynamically expanding specialized blocks. These blocks, optimized through a targeted hyperparameter search, focus primarily on task-specific learning.

A related yet distinct task is *contrastive learning*, which pairs positive and negative instances with corresponding anchors to enrich training (Chen et al., 2020; Ott et al., 2023). While continual learning with contrastive techniques is prevalent in semantic tasks (Ke et al., 2021), contrastive learning's application in continual learning for image classification tasks is limited (Cha et al., 2021). This limitation arises from the potential decrease in diversity of negative samples, affecting the task-specific loss, and the likelihood of the regularizer may hinder learning of new distinctive representations (Cha & Moon, 2023). Recognizing the importance of weighting both the classification and contrastive loss functions to achieve a balance between previous and new tasks, our approach utilizes a Bayesian learning-driven strategy (Kendall & Gal, 2017; Kendall et al., 2018). A notable challenge arises from the inconsistency of negative samples between classification and representation learning tasks (Liu et al., 2024). A Bayesian framework facilitates the inference of probabilities for contrastive model samples, eliminating the necessity for computationally intensive hyperparameter tuning searches (Hasanzadeh et al., 2022). Figure 1 shows the main idea of our approach, exemplary for GNSS interference classification. For each task, two additional classes are trained and the model weights are averaged with the weights from the previous task to circumvent catastrophic forgetting by adapting only the specialized blocks. We combine the cross-entropy (CE) with a contrastive learning (CL) loss and dynamically weigh them with Bayesian learning. This leads to an optimal feature representation with improved clusters.

The main contribution of this work is to propose a continual learning approach capable of adapting to novel classes. Our contributions are as follows: (1) We introduce BLCL, a continual learning method initially based on MEMO (Zhou et al., 2023b), which is founded on generalized and specialized blocks. In addition to this setting, we introduce dynamic specialized blocks by adding additional blocks as shown in Figure 2. (2) We combine the CE with a CL loss, incorporating class embedding distances in a prototypical manner (Snell et al., 2017). (3) We utilize a Bayesian learning-driven (Kendall et al., 2018) dynamic weighting mechanism for both loss functions, and (4) we conduct extensive evaluations on CIFAR-10, CIFAR-100 (Krizhevsky & Hinton, 2009), ImageNet100, and a GNSS (Ott et al., 2024) dataset to demonstrate that BLCL yields an enhanced feature representation, resulting in improved classification accuracy for image-related tasks.

## 2 Related Work

### 2.1 Continual/Class-Incremental Learning

Class-incremental learning (CIL) aims to continuously develop a combined classifier for all encountered classes. The primary challenge in CIL lies in catastrophic forgetting, where direct optimization of the network with new classes leads to the erasure of knowledge pertaining to former tasks, thereby resulting in irreversible performance degradation. The objective is to effectively mitigate catastrophic forgetting (Zhou et al., 2023a). Conventional ML models address CIL through dual tasks by updating only the model with a single new stage (Lewandowsky & Li, 1995; Kuzborskij et al., 2013; Da et al., 2014). Feature extraction methods (Donahua et al., 2014) maintain the set of shared network parameters ($\Theta_s$) and task-specific parameters for previously learned tasks ($\Theta_o$) unchanged. Here, only the outputs of one or more layers serve as features for the new task, employing randomly initialized task-specific parameters ($\Theta_n$). In finetune (Girshick et al., 2014) methods, $\Theta_o$ remains fixed, while $\Theta_s$ and $\Theta_n$ are optimized for the new task. Kirkpatrick et al. (2017) proposed elastic weight consolidation (EWC), which selectively decelerates learning on weights crucial for previous tasks. The method learning without forgetting (LwF), introduced by Li & Hoiem (2018), employs solely new task data to train the network while retaining the original capabilities. Rolnick et al. (2019) proposed CLEAR (continual learning with experience and replay) for multi-task reinforcement learning, leveraging off-policy learning and behavioral cloning from replay to enhance stability, as well as on-policy learning to preserve plasticity. Rebuffi et al. (2017) introduced iCaRL, which learns such that only the training data for a small number of classes needs to be present concurrently, allowing for progressive addition of new classes. iCaRL combines a classification loss with a distillation loss to replicate the scores stored in the previous step. Replay, introduced by Ratcliff (1997), investigates manipulations of the network within a multilayer model and various variants thereof. Dynamically expandable representation (DER) (Yan et al., 2021) considers limited memory and aim to achieve better stability-plasticity trade-off utilizing a dynamically expandable representation. Inspired by gradient boosting, FOSTER (Wang et al., 2022) dynamically expands new modules to fit the residual between the target and output of the original model. Yu et al. (2020) compensated the semantic drift of features for embedding networks without exemplars. PASS (Zhu et al., 2021) and Zhu et al. (2022) mainly focus on previous class prototypes and updating the backbone model.

Conventional CIL methods typically prioritize representative exemplars from previous classes to mitigate forgetting (Luo et al., 2023). However, recent investigations indicate that preserving models from history can significantly enhance performance. Certain applications necessitate memory-efficient architectures. However, state-of-the-art CIL methods are not compared with regard to their memory budget. The memory-efficient expandable model (MEMO) (Zhou et al., 2023b) addresses this gap by considering accuracy and memory size, analyzing the impact of different network layers. Zhou et al. (2023b) discovered that shallow and deep layers exhibit distinct characteristics in the context of CIL. MEMO extends specialized layers based on shared generalized representations, efficiently extracting diverse representations at modest cost while maintaining representative exemplars. We adopt MEMO as our methodological baseline. However, we observe that results can be further enhanced by incorporating dynamic specialized blocks (the last layers in Figure 1 for which the weights are averaged with the previous task). In our experiments, we compare the performance of Finetune, EWC, LwF, iCarl, Replay, MEMO, DER, and FOSTER. For a comprehensive overview of methodologies, refer to Zhou et al. (2023a); Lange et al. (2022); Masana et al. (2023); Mai et al. (2022).

### 2.2 Bayesian Contrastive Learning

Contrastive learning establishes pairs of positive and negative instances along corresponding anchors to enhance training (Chen et al., 2020). This approach has found extensive application across various domains including multi-modal learning (Wei et al., 2016; Rasiwasia et al., 2010; Huang et al., 2020), domain adaptation (Thota & Leontidis, 2021), and semantic learning tasks (Ke et al., 2021; Sarafianos et al., 2019). The contrastive learning paradigm has been extended to triplet learning (Ott et al., 2023; Schroff et al., 2015), which utilizes a positive sample, and quadruplet learning (Ott et al., 2024; Chen et al., 2017), which employs a similar sample. The main focus within this field resolves around the optimal selection of pairs to augment training. However, the primary focus of our study resides in determining the optimal weighting between classification and contrastive loss functions.

Cha et al. (2021) demonstrated that representations learned contrastively exhibit greater robustness against catastrophic forgetting compared to those trained using the CE loss. Their approach integrates an asymmetric variant of supervised contrastive loss, mitigating model overfitting to a limited number of previous task samples by employing a dynamic architecture. However, contrastive learning requires a large number of diverse negative samples to be effective. On smaller datasets (as the CIFAR-10 and GNSS datasets), this lack of diversity can degrade performance (Cha et al., 2021). Similarly, Sy-CON (Cha & Moon, 2023) also integrates symmetric contrastive loss. Hasanzadeh et al. (2022) utilized distributional representations to provide uncertainty estimates in downstream graph analytic tasks. Lin et al. (2023) employ Monte Carlo dropout on skeleton data for action recognition, generating pairwise samples for model robustness. Liu et al. (2024) introduced a Bayesian contrastive loss to mitigate false negatives and mine hard negative samples, aligning the semantics of negative samples across tasks. Instead, our approach leverages Bayesian learning as proposed by Kendall et al. (2018) for weighting the CE and CL functions.

## 3 Methodology

This section provides a detailed description of our methodology. In Section 3.1, we mathematically formulate the problem of CIL. Subsequently, in Section 3.2, we provide an overview of our proposed method. We introduce a contrastive loss formulation and Bayesian learning-driven weighting strategy in Section 3.3. We describe data augmentation techniques in the Appendix A.1.

### 3.1 Problem Formulation

We consider $T$ sequentially presented training tasks denoted as $\{\mathcal{D}^1, \mathcal{D}^2, \cdots, \mathcal{D}^T\}$, where each $\mathcal{D}^t = \{(\mathbf{X}_i^t, y_i^t)\}_{i=1}^{n_t}$ represents the $t$-th incremental step containing $n_t$ training instances of images. $\mathbf{X}_i^t$ signifies a sample belonging to class $y_i \in Y_t$, with $Y_t$ representing the label space of task $t$. During the training of task $t$, only data from $\mathcal{D}^t$ is available. The objective is to continuously develop a classification model encompassing all classes encountered thus far. Subsequent to each task, the model is evaluated for all observed classes $\mathcal{Y}_t = Y_1 \cup \cdots \cup Y_t$ (Zhou et al., 2023a; Rebuffi et al., 2017). Typically, classes do not overlap between different tasks. However, in certain scenarios, previously encountered classes may reappear in subsequent tasks, constituting what is termed as blurry CIL (Xie et al., 2022; Koh et al., 2022; Bang et al., 2021; 2022). For the purpose of this paper, it is noted that the classes are overlapping as we have a fixed number of representative instances from the previous task classes, so-called *exemplars*.

### 3.2 Method Overview

Our methodology is outlined in Figure 2, designated as BLCL. Our BLCL method is built upon the MEMO architecture (Zhou et al., 2023b), which serves as the baseline. Our key contribution lies in the integration of a Bayesian framework to dynamically balance loss functions. Initially, we trained 110 vision encoder models from Hugging Face (Wightman, 2019) on the GNSS dataset, identifying the ResNet model as the optimal backbone (see Appendix A.2). The ResNet model, proposed by He et al. (2016), specifically ResNet18 or ResNet32, comprises both generalized and specialized components. The generalized component extracts features from the image dataset, featuring 13 convolutional layers, and is succeeded by one or more (up to four) additional layers. The specialized component consists of blocks whose weights are averaged with the weights from the previous task. These specialized layers contain different numbers of blocks $l$, adapted to the dataset at hand, are denoted as $\text{BLCL}_l$, where each block comprises of two convolutional layers and two batch normalizations. The output dimension of the convolutional layers is set to 512. We employ average pooling and a fully connected (FC) layer, and we train utilizing the CE loss $\mathcal{L}_{\text{CE}}$. The training process involves freezing the specialized component from previous tasks while updating the generalized and specialized component with classes pertinent to the specific tasks. Moreover, we average the weights of the specialized component from previous tasks. To maintain adaptability to varying complexities of new classes across tasks, we dynamically adjust each task's layer structure, denoted as $\text{BLCL}_{l[s]}$ (as depicted in the top right of Figure 2). For example, the notation $\text{BLCL}_{8[1,1,2,2]}$ represents a network architecture consisting of eight specialized blocks, where the final layers are arranged in the sequence 1, 1, 2, 2 for each task. This adaptation involves the removal of either one convolutional layer or an entire block from the specialized

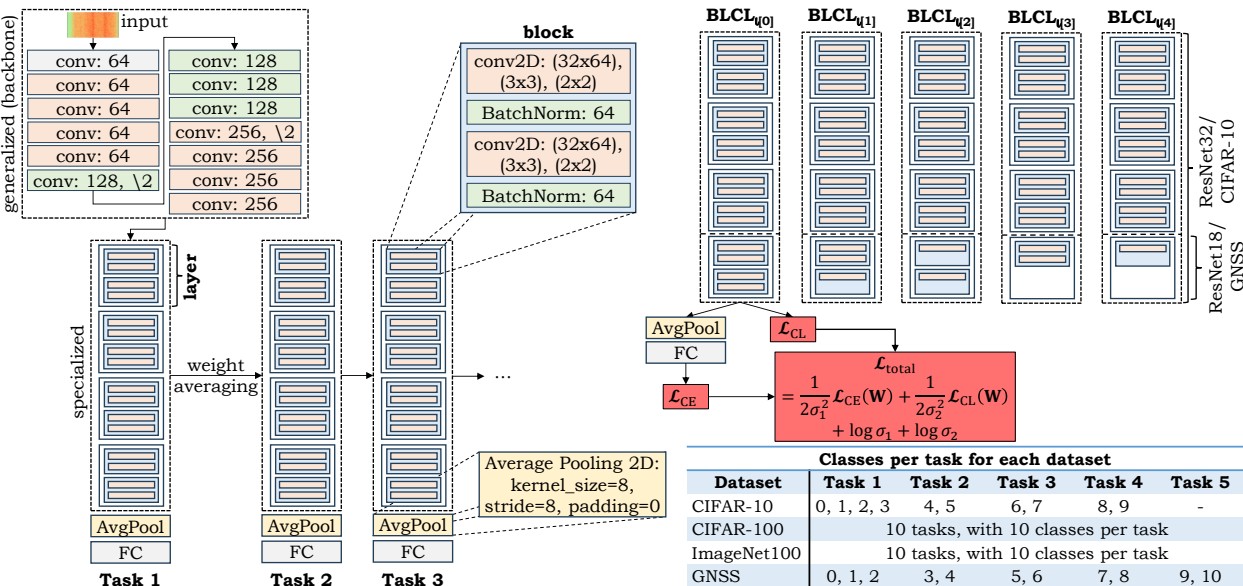

Figure 2: Overview of our BLCL method. After the generalized component of ResNet18 or ResNet32 (He et al., 2016), follow up to eight blocks ($l = 8$) for the CIFAR (Krizhevsky & Hinton, 2009) datasets and two blocks ($l = 2$) for the GNSS (Ott et al., 2024) dataset in their respective specialized blocks. After each task the weights are averaged with the weights of the previous tasks. We propose a dynamic architecture with a different number of blocks and convolutional layers, indicated by $\text{BLCL}_{l[s]}$. After the FC layer, follows a CE loss in combination with a Bayesian learning-driven weighting of the contrastive loss (CL).

model's layer configuration, catering to the evolving requirements of each task. Furthermore, to facilitate a fair comparison with other state-of-the-art methods and to compensate for the additional blocks, we maintain a constant number of exemplars, fixed at 2,000 samples, throughout the training process for all datasets. See the Appendix A.3, for the dynamic adjustment of the layer structure.

### 3.3 Loss Functions

The lower-level representation of the CIL model is continuously adapted with the addition of each new class, thereby potentially undergoing significant changes. The primary objective is to incrementally enhance class prototypes, thereby reducing intra-class distances while simultaneously increasing inter-class distances. To facilitate this objective, we introduce a contrastive loss term, denoted as $\mathcal{L}_{\text{CL}}$. Achieving an optimal balance between the CE and CL loss functions is crucial. Hence, we propose an automated weighting mechanism utilizing Bayesian learning techniques.

**Contrastive Loss.** Pairwise learning is characterized by the utilization of pairs featuring distinct labels, serving to enrich the training process by enforcing a margin between pairs of images belonging to different identities. Consequently, the feature embedding for a specific label lives on a manifold while ensuring sufficient discriminability, i.e., distance, from other identities (Ott et al., 2023; Do et al., 2019). CL, on the other hand, is a methodology aimed at training models to learn representations by minimizing the distance between positive pairs and maximizing the distance between negative pairs within a latent space. We define feature embeddings $f(\mathbf{X}) \in \mathbb{R}^{q \times h}$ to map the image input into a feature space $\mathbb{R}^{q \times h}$, where $f$ represents the output of the last convolutional layer with a size of 512 from the specialized model, and $q \times h$ denotes the dimensionality of the layer output. Consequently, we define an *anchor* sample $\mathbf{X}_i^a$ corresponding to a specific label, a *positive* sample $\mathbf{X}_i^p$ from the same label, and a *negative* sample $\mathbf{X}_i^n$ drawn from a different label. For all training samples $\big(f(\mathbf{X}_i^a), f(\mathbf{X}_i^p), f(\mathbf{X}_i^n)\big) \in \Phi$, our objective is to fulfill the following inequality:

$$d\big(f(\mathbf{X}_i^a), f(\mathbf{X}_i^p)\big) + \alpha < d\big(f(\mathbf{X}_i^a), f(\mathbf{X}_i^n)\big), \tag{1}$$

where $d$ is a distance function, specifically the cosine distance $\mathrm{CD}(\mathbf{x}_1, \mathbf{x}_2) = 1 - \frac{\mathbf{x}_1 \cdot \mathbf{x}_2}{||\mathbf{x}_1|| \cdot ||\mathbf{x}_2||}$ computed between two non-zero vectors $\mathbf{x}_1, \mathbf{x}_2 \in \mathbb{R}^n$ of size $n$, $\alpha = 1$ denotes a margin differentiating positive and negative pairs, and $\Phi$ is the set of all possible pairs within the training dataset, where $N$ is the number of pairs. For each batch of size $bs$, we select all conceivable pairs and increase the batch size to $\frac{bs(bs+1)}{2}$.

---

**Algorithm 1** BLCL: Algorithm for Class-Incremental Learning (Python & PyTorch like code)

---

1: **Input:** $T$ = task sequence, $t$ = current task, $M$ = model, $\tau$ = similarity threshold,
2:      $E$ = number of exemplars, $N$ = number of epochs, $\alpha$ = learning rate
3: **for** each $t$ in $T$ **do**
4:      dataset = `load_task_data`($t$)
5:      class_similarity = `class_similarity`($t$, `previous_tasks`)          ▷ Based on cosine similarity
6:      **if** class_similarity $> \tau$ **then**
7:          `reduce_layers`($M$)                          ▷ Reduce layers in the last specialized block
8:      **else**
9:          `retain_layers`($M$)                        ▷ Keep all layers from the last specialized block
10:      **end if**
11:      optimizer = `initialize_optimizer`($M$, $\sigma_1$, $\sigma_2$, $\alpha$)                          ▷ Adam optimizer
12:      `freeze_previous_specialized_layers`($M$, $\{t-1, \ldots, 1\}$)
13:      `enable_training_for_generalized_and_specialized`($M$, $t$)
14:      **for** epoch = 1 **to** $N$ **do**
15:          **for** (X, Y) in dataset **do**
16:              softmax, emb = `model(X)`                          ▷ Embeddings of laster layer
17:              $\mathcal{L}_{\mathrm{CE}}$ = `cross_entropy_loss(softmax, Y)`
18:              pos_pair, neg_pair = `form_pairs(emb)`   ▷ Select positive and negative pairs for each batch
19:              $\mathcal{L}_{\mathrm{CL}}$ = `contrastive_loss(emb, pos_pair, neg_pair)`
20:              loss, $\sigma_1$, $\sigma_2$ = `Bayesian_weighting`($\mathcal{L}_{\mathrm{CE}}$, $\mathcal{L}_{\mathrm{CL}}$, $\sigma_1$, $\sigma_2$)          ▷ Update deviation parameters
21:              optimizer.zero_grad( )
22:              loss.backward( )
23:              optimizer.step( )
24:          **end for**
25:      **end for**
26:      $M_t$ = `average_specialized_weights`($M_t$, $M_{t-1}$)
27:      $E$ = `update_exemplars`($t$, $E$)                          ▷ Ensures that the total exemplars are equal to $E$
28: **end for**
29: **Return:** trained $M$

---

**Bayesian Learning.** Typically, naïve methods employ a weighted combination of both loss functions to compute the total loss, denoted as $\mathcal{L}_{\mathrm{total}} = w_1 \mathcal{L}_{\mathrm{CE}} + w_2 \mathcal{L}_{\mathrm{CL}}$. However, the model performance is extremely sensitive to the selection of weights $w_i$ (Groenendijk et al., 2021). Balancing CE and CL losses is essential because they capture complementary aspects of learning: CE ensures class-specific discrimination, while CL enhances feature representations. Optimizing only the CL may lead to poor decision boundaries and degraded classification performance. Dynamic weighting allows the model to adapt throughout training, emphasizing representation learning initially and refining decision boundaries later for better generalization. In our approach, we adopt a strategy wherein we concurrently optimize both objectives using homoscedastic task uncertainty, as defined by Kendall & Gal (2017); Kendall et al. (2018). This homoscedastic uncertainty pertains to the task-dependent aleatoric uncertainty, which remains invariant across input data but varies across distinct tasks (Klaß et al., 2022). Let $\mathrm{f}^{\mathbf{W}}(\mathbf{X})$ denote the output of a neural network with weights $\mathbf{W}$ on input $\mathbf{X}$. In scenarios involving multiple model outputs, we factorize over the outputs

$$p(\mathbf{y}_1, \ldots, \mathbf{y}_K | \mathrm{f}^{\mathbf{W}}(\mathbf{X})) = p(\mathbf{y}_1 | \mathrm{f}^{\mathbf{W}}(\mathbf{X})) \cdots p(\mathbf{y}_K | \mathrm{f}^{\mathbf{W}}(\mathbf{X})), \tag{2}$$

with $K$ model outputs $\mathbf{y}_1, \ldots, \mathbf{y}_K$ (Kendall et al., 2018). In a classification task, we sample from a probability vector from a scaled softmax function's output $p(\mathbf{y} | \mathrm{f}^{\mathbf{W}}(\mathbf{X}), \sigma) = \mathrm{softmax}(\frac{1}{\sigma^2} \mathrm{f}^{\mathbf{W}}(\mathbf{X}))$. The log likelihood is

then defined as

$$\log p\big(\mathbf{y} = c | \mathrm{f}^{\mathbf{W}}(\mathbf{X}), \sigma\big) = \frac{1}{\sigma^2} \mathrm{f}_c^{\mathbf{W}}(\mathbf{X}) - \log \sum_{c'} \exp\big(\frac{1}{\sigma^2} \mathrm{f}_c^{\mathbf{W}}(\mathbf{X})\big), \tag{3}$$

where $\mathrm{f}_c^{\mathbf{W}}(\mathbf{X})$ is the $c$-th element of $\mathrm{f}^{\mathbf{W}}(\mathbf{X})$. In a regression task, we maximize the log likelihood of the model as

$$\log p\big(\mathbf{y} | \mathrm{f}^{\mathbf{W}}(\mathbf{X}), \sigma\big) \propto -\frac{1}{2\sigma^2} ||\mathbf{y} - \mathrm{f}^{\mathbf{W}}(\mathbf{X})||^2 - \log \sigma, \tag{4}$$

for a Gaussian likelihood $p\big(\mathbf{y} | \mathrm{f}^{\mathbf{W}}(\mathbf{X})\big) = \mathcal{N}\big(\mathrm{f}^{\mathbf{W}}(\mathbf{X}), \sigma^2\big)$, where $\sigma$ is the model's observation noise parameter (Kendall et al., 2018). In our case, our model output is composed of two vectors, a discrete output $\mathbf{y}_1$ for the CE loss and continuous output $\mathbf{y}_2$ for the CL loss, which leads to the total minimization objective:

$$\begin{aligned}
\mathcal{L}(\mathbf{W}, \mathbf{X}, \sigma_1, \sigma_2) &= -\log p\big(\mathbf{y}_1 = c, \mathbf{y}_2 | \mathrm{f}^{\mathbf{W}}(\mathbf{X})\big) = -\mathrm{softmax}\big(\mathbf{y}_1 = c; \mathrm{f}^{\mathbf{W}}(\mathbf{X}), \sigma_1\big) + \log \mathcal{N}\big(\mathbf{y}_2; \mathrm{f}^{\mathbf{W}}(\mathbf{X}), \sigma_2^2\big) \\
&= -\log p\big(\mathbf{y}_1 = c | \mathrm{f}^{\mathbf{W}}(\mathbf{X}), \sigma_1\big) + \frac{1}{2\sigma_2^2} ||\mathbf{y}_2 - \mathrm{f}^{\mathbf{W}}(\mathbf{X})||^2 + \log \sigma_2 \\
&\approx \frac{1}{2\sigma_1^2} \mathcal{L}_{\mathrm{CE}}(\mathbf{W}, \mathbf{X}) + \frac{1}{2\sigma_2^2} \mathcal{L}_{\mathrm{CL}}(\mathbf{W}, \mathbf{X}) + \log \sigma_1 + \log \sigma_2.
\end{aligned} \tag{5}$$

The final objective is to minimize Equation 5 with respect to $\sigma_1$ and $\sigma_2$, thereby learning the relative weights of the losses $\mathcal{L}_1(\mathbf{W})$ and $\mathcal{L}_2(\mathbf{W})$ in an adaptive manner (Kendall & Gal, 2017; Kendall et al., 2018). The hyperparameters $\sigma_1$ and $\sigma_2$ are trainable parameters, initially set to one, indicating equal weighting for both. The Appendix A.4 provides an illustrative demonstration of the weighting mechanism applied to both loss functions for all tasks on the CIFAR-10, CIFAR-100, ImageNet100, and GNSS datasets. As the noise parameters $\sigma_1$ and $\sigma_2$ increase – where $\sigma_1$ corresponds to the variable $\mathbf{y}_1$ and $\sigma_2$ to $\mathbf{y}_2$ – the weight assigned to $\mathcal{L}_{\mathrm{CE}}$ and $\mathcal{L}_{\mathrm{CL}}$, respectively, decreases. Such an adjustment serves as a regularizer for the noise term. This trend is observable at the early period of each task, as new classes introduce more noise, and hence, the weighting of both loss functions decreases. In general, both loss functions have converged after 200 epochs.

The function `Bayesian_weighting` dynamically balances the contributions of the CE loss ($L_{CE}$) and the CL loss ($L_{CL}$) using Bayesian uncertainty estimation. It introduces two deviation parameters, $\sigma_1$ and $\sigma_2$, which represent the uncertainty associated with each loss term. The function computes weighting factors as $1/(2\sigma_1^2)$ and $1/(2\sigma_2^2)$, ensuring that losses with higher uncertainty contribute less to the total loss. Additionally, a regularization term, $\log(\sigma_1) + \log(\sigma_2)$, is included to prevent the deviation parameters from collapsing to zero. The final weighted loss is then returned along with the updated values of $\sigma_1$ and $\sigma_2$, which are adjusted using gradient-based optimization. This adaptive weighting mechanism ensures that the model prioritizes more reliable loss terms during training, improving robustness in an incremental learning setting.

The function `update_exemplars` is responsible for maintaining and updating the exemplar memory used in CIL. Given the current task index $t$ and the existing exemplar set $E = 2,000$, the function determines the number of exemplars per class as $\frac{E}{k}$, where $k$ is the number of classes at the current stage. It then selects a subset of training samples for each class using a predefined strategy called Herding. The goal is to retain the most representative exemplars while adhering to memory constraints.

## 4  Experiments

For fair comparability, we conduct experiments on four distinct image datasets: the CIFAR-10, CIFAR-100, and ImageNet100 datasets and a GNSS-based dataset. Consequently, we encounter a diverse array of applications pertinent to continual learning, specifically within the realm of image classification tasks.

### 4.1  Image Classification: CIFAR-10, CIFAR-100, & ImageNet100

The CIFAR-10 (Krizhevsky & Hinton, 2009) dataset consists of 60,000 colour images of size $32 \times 32$. The goal is to classify 10 distinct classes (i.e., airplane, automobile, bird, cat, deer, dog, etc.), with 6,000 images per class. The dataset is split into 50,000 training and 10,000 test images. We train four tasks: task 1 consists of the classes 0, 1, 2, and 3, task 2 consists of the classes 4 and 5, task 3 consists of the classes

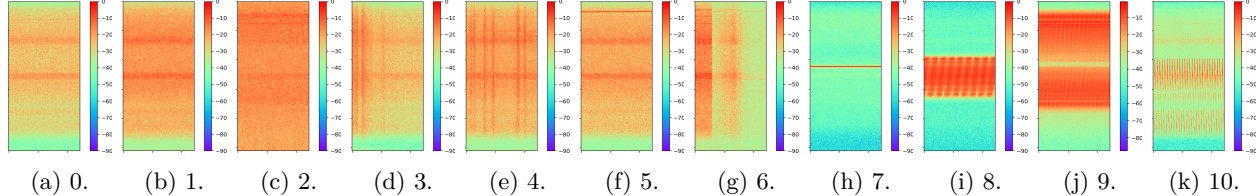

| (a) 0. | (b) 1. | (c) 2. | (d) 3. | (e) 4. | (f) 5. | (g) 6. | (h) 7. | (i) 8. | (j) 9. | (k) 10. |

Figure 3: Exemplary spectrogram samples without interference (classes 0 to 2) and with interference (classes 3 to 10) between intensity [0, -90] with logarithmic scale.

6 and 7, and task 4 consists of the classes 8 and 9. The CIFAR-100 (Krizhevsky & Hinton, 2009) dataset has 100 classes containing 600 images each, and hence, we train 10 classes per task with 10 tasks in total. ImageNet100 is a subset of the larger ImageNet dataset, containing 100 carefully selected classes with 1,300 images per class, maintaining a balanced distribution. It is commonly used for benchmarking due to its reduced computational requirements while preserving the diversity of ImageNet (Deng et al., 2009).

## 4.2 Interference Classification: GNSS Snapshots

The GNSS-based (Ott et al., 2024) dataset contains short, wideband snapshots in both E1 and E6 GNSS bands. The dataset was captured at a bridge over a highway. The setup records 20 ms raw IQ (in-phase and quadrature-phase) snapshots triggered from the energy with a sample rate of 62.5 MHz, an analog bandwidth of 50 MHz and an 8 bit bit-width. Further technical details can be found in (Brieger et al., 2022; Heublein et al., 2024a; 2025). Figure 3 shows exemplary snapshots of the spectrogram. Manual labeling of these snapshots has resulted in 11 classes: Classes 0 to 2 represent samples with no interferences, distinguished by variations in background intensity, while classes 3 to 10 contain different interferences such as pulsed, noise, tone, and multitone. For instance, Figure 3k showcases a snapshot containing a potential chirp jammer type. The dataset's imbalance of 197,574 samples for non-interference classes and between 9 to 79 samples per interference class, emphasizing the under-representation of positive class labels. The goal is to adapt to new interference types with continual learning. The challenge lies in adapting to positive class labels with only a limited number of samples available. We partition the dataset into a 64% training set, 16% validation set, and a 20% test set split (balanced over the classes). We train five tasks: task 1 consists of the classes 0, 1, and 2, task 2 consists of the classes 3 and 4, task 3 consists of the classes 5 and 6, task 4 consists of the classes 7 and 8, and task 5 consists of the classes 9 and 10.

# 5 Evaluation

Initially, we present the results obtained through our BLCL approach, in comparison with state-of-the-art methods, on both the CIFAR, the ImageNet, and the GNSS datasets. Subsequently, we conduct an analysis involving confusion matrices, model embeddings, cluster analysis, and parameter counts. Throughout, we provide the accuracy achieved after each task, the average accuracy over all tasks ($\overline{\text{Acc.}}$), as well as the F1-score and F2-score. For our hardware and training setup, see the Appendix A.5.

Table 1: Evaluation results for CIFAR-100.

| Method | Final | | | | |
| --- | --- | --- | --- | --- | --- |
| | Acc. | F1 | F2 | DB | CH |
| Finetune | 26.41 | 0.01 | 0.03 | 4.34 | 68 |
| EWC (Kirkpatrick et al., 2017) | 35.81 | 0.20 | 0.20 | 3.11 | 77 |
| LwF (Li & Hoiem, 2018) | 43.62 | 0.15 | 0.17 | 2.99 | 72 |
| iCarl (Rebuffi et al., 2017) | 54.83 | 0.35 | 0.33 | 2.67 | 63 |
| Replay (Ratcliff, 1997) | 52.90 | 0.34 | 0.31 | 2.77 | 68 |
| MEMO (Zhou et al., 2023b), w/o | 68.86 | 0.57 | 0.57 | 3.55 | 55 |
| DER (Yan et al., 2021) | 67.17 | 0.58 | **0.58** | 3.37 | 61 |
| FOSTER (Wang et al., 2022) | 63.51 | 0.50 | 0.50 | 3.52 | 74 |
| BLCL$_{8[s]}$, CE | 70.14 | 0.57 | 0.57 | 3.58 | 57 |
| BLCL$_{8[s]}$, CL | 5.24 | 0.01 | 0.02 | 5.32 | 76 |
| BLCL$_{8[s]}$, w/ | **70.43** | **0.58** | **0.58** | **2.59** | **102** |
| $\pm$ | 0.50 | 0.007 | 0.007 | 0.019 | 1.15 |

**Evaluation Results.** The evaluation results for the CIFAR datasets are summarized in Table 1 and Table 2, while those for the GNSS dataset are presented in Table 3. On the CIFAR-10 dataset, MEMO and DER achieve the highest classification results for each task, with the highest average accuracy of 84.78%, respectively 86.57%. iCarl also demonstrates strong performance with an average accuracy of 84.47%. However, MEMO with weight averaging slightly decreases results to 84.78%, leading us to train BLCL without

Table 2: Evaluation results for baseline methods and our Bayesian learning-driven (B) contrastive loss functions for the CIFAR-10 (Krizhevsky & Hinton, 2009) dataset. We indicate the weighting with (w/) and without (w/o) averaging. Underlined is the best baseline method and **bold** is the best method in total. We present the final Davies-Bouldin (DB) (↓) and Caliński-Harbasz (CH) (↑) scores.

| Method | Loss/ Weighting | Task 1 Acc. | Task 2 Acc. | Task 3 Acc. | Task 4 Acc. | Final Acc. | F1 | F2 | DB | CH |
|---|---|---|---|---|---|---|---|---|---|---|
| Finetune | CE | 95.20 | 32.08 | 24.80 | 19.80 | 43.22 | 0.067 | 0.110 | 7.29 | 1,716 |
| EWC (Kirkpatrick et al., 2017) | | 95.20 | 54.39 | 47.39 | 36.10 | 58.24 | 0.303 | 0.324 | 2.12 | 1,365 |
| LwF (Li & Hoiem, 2018) | | 95.48 | 63.55 | 62.80 | 56.89 | 69.68 | 0.556 | 0.557 | 2.14 | 848 |
| iCarl (Rebuffi et al., 2017) | | 95.35 | 81.85 | 80.12 | 79.57 | 84.47 | 0.792 | 0.791 | 1.28 | 1,907 |
| Replay (Ratcliff, 1997) | | 95.35 | 78.95 | 76.69 | 76.72 | 81.93 | 0.766 | 0.764 | 1.18 | 2,327 |
| MEMO (Zhou et al., 2023b), w/o | CE | **96.00** | 80.68 | 80.79 | 81.68 | 84.78 | 0.810 | 0.812 | 1.78 | 2,327 |
| DER (Yan et al., 2021) | | 95.32 | 84.23 | 83.70 | 83.48 | 86.15 | 0.833 | 0.834 | 1.38 | 1,822 |
| FOSTER (Wang et al., 2022) | | 95.95 | 69.58 | 66.90 | 66.86 | 74.82 | 0.637 | 0.645 | 2.20 | 1,225 |
| BLCL$_{8[1,1,2,2]}$, w/o | CE | 92.82 | 82.47 | 82.18 | 82.13 | 84.9 | 0.816 | 0.817 | 1.74 | 1,353 |
| BLCL$_{8[1,1,2,2]}$, w/o | CL | 50.55 | 36.98 | 28.77 | 21.66 | 34.49 | 0.068 | 0.108 | 7.89 | 787 |
| BLCL$_{8[1,1,1,2]}$, w/o | CE, B, CL | 95.58 | **84.50** | 82.55 | **83.62** | 86.56 | **0.834** | **0.834** | **0.79** | **5,085** |
| ± | | 0.200 | 0.422 | 0.326 | 0.508 | 0.246 | 0.005 | 0.005 | 0.010 | 81.34 |
| BLCL$_{8[1,1,2,2]}$, w/o | CE, B, CL | 95.58 | **84.50** | 83.45 | 82.76 | **86.57** | 0.824 | 0.825 | 0.81 | 4,688 |
| ± | | 0.200 | 0.265 | 0.373 | 0.194 | 0.099 | 0.002 | 0.002 | 0.007 | 99.107 |

Table 3: Evaluation results for the GNSS (Ott et al., 2024) dataset.

| Method | Loss/ Weighting | Task 1 Acc. | Task 2 Acc. | Task 3 Acc. | Task 4 Acc. | Task 5 Acc. | Final Acc. | F1 | F2 | DB | CH |
|---|---|---|---|---|---|---|---|---|---|---|---|
| Finetune | CE | 96.06 | 0.04 | 0.04 | 0.06 | 0.02 | 19.24 | 0.033 | 0.048 | 1.48 | 7,958 |
| EWC (Kirkpatrick et al., 2017) | | 95.99 | 0.49 | 0.04 | 0.05 | 0.04 | 19.32 | 0.071 | 0.072 | 1.34 | 4,799 |
| LwF (Li & Hoiem, 2018) | | 96.40 | 52.39 | 17.49 | 3.50 | 2.52 | 34.46 | 0.026 | 0.037 | 1.97 | 4,565 |
| iCarl (Rebuffi et al., 2017) | | 94.75 | 95.43 | 95.23 | 95.14 | 95.14 | 95.14 | 0.601 | 0.651 | 0.81 | 29,076 |
| Replay (Ratcliff, 1997) | | 94.55 | 95.97 | 95.49 | 95.50 | 95.75 | 95.45 | 0.594 | 0.661 | 0.96 | 27,622 |
| MEMO (Zhou et al., 2023b), w/o | CE | 95.95 | 95.41 | 94.99 | 95.14 | 95.19 | 95.34 | 0.579 | 0.651 | 1.15 | 14,066 |
| DER (Yan et al., 2021) | | 96.08 | 95.73 | 95.61 | 95.67 | 95.78 | 95.77 | 0.562 | 0.642 | 1.15 | 8,828 |
| FOSTER (Wang et al., 2022) | | 96.24 | 93.67 | 92.30 | 91.09 | 93.53 | 93.36 | 0.401 | 0.479 | 1.53 | 10,402 |
| BLCL$_{2[2,2,2,2,2]}$, w/ | CE | 96.39 | 95.87 | 94.89 | 94.37 | 94.26 | 95.35 | 0.551 | 0.600 | 1.26 | 12,553 |
| BLCL$_{2[2,2,2,2,2]}$, w/ | CL | 66.61 | 12.1 | 8.29 | 3.11 | 2.28 | 18.478 | 0.024 | 0.024 | 1.98 | 5,567 |
| BLCL$_{2[1,1,1,1,1]}$, w/ | CE, 0.9, CL | 96.03 | 95.99 | 95.54 | 95.61 | 95.17 | 95.67 | **0.619** | **0.678** | 1.04 | 15,868 |
| BLCL$_{2[1,1,1,1,1]}$, w/ | CE, B, CL | 95.12 | 95.35 | 94.58 | 94.39 | 93.74 | 94.63 | 0.547 | 0.596 | 0.79 | 98,877 |
| BLCL$_{2[2,2,2,2,2]}$, w/ | CE, 0.9, CL | 96.21 | 95.37 | 95.13 | 95.44 | **96.25** | 95.68 | 0.612 | 0.648 | 0.94 | 28,598 |
| BLCL$_{2[2,2,2,2,2]}$, w/ | CE, B, CL | **96.74** | **96.16** | **95.75** | **95.96** | 96.18 | **96.16** | 0.601 | 0.662 | **0.78** | **163,420** |
| ± | | 0.135 | 0.252 | 0.123 | 0.250 | 0.298 | 0.244 | 0.021 | 0.023 | 0.007 | 386.05 |

weight averaging. Figure 4 shows results for different BLCL$_{l[s]}$ architectures. Increasing the number of layers in the specialized component from four (84.78%) to six (84.96%) and eight (85.02%) results in improved accuracy. Consequently, follow-up experiments are conducted with the BLCL$_{8[s]}$ model. Evaluation of dynamic specialized components reveals that smaller blocks are preferable for the first task, as BLCL$_{8[3,...]}$ and

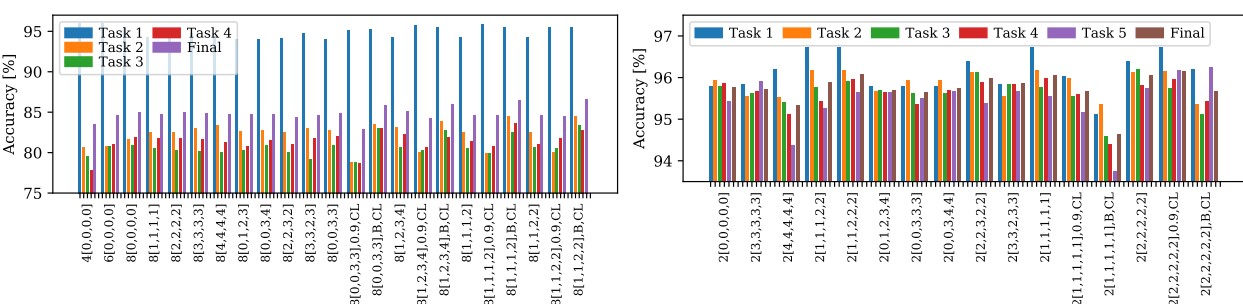

Figure 4: Evaluation of BLCL$_{l[s]}$ on the CIFAR-10 (a) and GNSS (b) datasets.

Table 4: Evaluation results for the ImageNet100 (Deng et al., 2009) dataset.

| Method | Task 1 Acc. | Task 2 Acc. | Task 3 Acc. | Task 4 Acc. | Task 5 Acc. | Task 6 Acc. | Task 7 Acc. | Task 8 Acc. | Task 9 Acc. | Task 10 Acc. | Final Acc. | F1 | F2 | DB | CH |
|---|---|---|---|---|---|---|---|---|---|---|---|---|---|---|---|
| Finetune | 79.60 | 44.30 | 25.07 | 18.50 | 15.32 | 14.90 | 13.29 | 10.88 | 9.36 | 9.20 | 24.04 | 0.02 | 0.38 | 3.99 | 11.34 |
| EWC | 85.20 | 47.30 | 35.13 | 26.25 | 21.20 | 17.60 | 14.20 | 14.25 | 13.93 | 10.96 | 28.60 | 0.06 | 0.07 | 2.74 | 26.87 |
| LwF | 87.00 | 75.40 | 66.40 | 52.20 | 43.24 | 36.23 | 31.06 | 30.52 | 28.91 | 23.24 | 47.42 | 0.18 | 0.19 | 3.24 | 17.18 |
| iCarl | 86.00 | 86.20 | 82.13 | 78.60 | 77.72 | 71.03 | 63.46 | 59.08 | 55.73 | 54.26 | 71.42 | 0.55 | 0.55 | 2.39 | 48.57 |
| Replay | 86.00 | 85.30 | 81.00 | 78.65 | 77.92 | 70.60 | 63.03 | 59.58 | 56.20 | 54.89 | 71.22 | 0.56 | 0.55 | 2.47 | 43.18 |
| MEMO | 89.40 | **88.50** | 84.20 | 83.60 | 82.12 | **79.30** | **75.23** | **75.00** | **72.00** | **71.00** | **81.04** | 0.71 | 0.71 | 3.03 | 37.86 |
| DER | 80.20 | 81.90 | 76.33 | 73.40 | 70.40 | 69.80 | 69.23 | 69.92 | 69.78 | 67.62 | 72.86 | **0.72** | **0.72** | 2.47 | 49.29 |
| FOSTER | 81.60 | 81.6 | 71.47 | 67.65 | 62.40 | 61.10 | 62.26 | 61.98 | 60.51 | 60.46 | 69.10 | 0.60 | 0.60 | 3.04 | 46.11 |
| $BLCL_{2[0,..,0]}$ | **91.80** | 88.30 | **85.40** | **85.30** | **82.92** | 79.17 | 74.43 | 72.55 | 68.78 | 68.60 | 79.73 | 0.68 | 0.68 | **2.28** | **59.10** |

(a) MEMO.  (b) MEMO.  (c) MEMO.  (d) MEMO.

(e) $BLCL_{8[1,1,2,2]}$.  (f) $BLCL_{8[s]}$.  (g) $BLCL_{2[2,2,2,2,2]}$.  (h) $BLCL_{2[0,..,0]}$.

Figure 5: Confusion matrices for the CIFAR-10 (a and e), CIFAR-100 (b and f), GNSS (c and g), and ImageNet100 (d and h) datasets. For confusion matrices of all methods, refer to the appendix.

$BLCL_{8[4,...]}$ outperform $BLCL_{8[0,...]}$. Larger layers for tasks 3 and 4 further improve results; for instance, $BLCL_{8[1,2,3,4]}$ achieves 85.11%. However, simply adding the contrastive loss with a naïve weighting of 0.9 decreases results, motivating the adoption of automatic Bayesian weighting. For all the architectures tested, including $BLCL_{8[1,1,1,2]}$, $BLCL_{8[1,1,2,2]}$, $BLCL_{8[0,0,3,3]}$, and $BLCL_{8[1,2,3,4]}$, our Bayesian loss outperforms the baseline model. Consequently, our method achieves 86.57% with high robustness across all tasks, significantly outperforming MEMO with 84.78% and DER with 86.15%. On the CIFAR-100 dataset, $BLCL_{8[s]}$ with $s = 0$ for all the 10 tasks, achieves 70.43% outperforming all state-of-the-art methods. On the GNSS dataset, Replay and DER achieve the highest accuracy of 95.45%, respectively 95.77%, although the F2-score of 0.661 is low due to dataset underrepresentation. We outperform MEMO (95.34% average accuracy) with weight averaging, achieving an accuracy of 95.67%. Combining this with the contrastive loss further increases accuracy to 96.16%. The F2-score can be improved to 0.678 by reducing the false negative rate. In Table 3, the Bayesian approach underperformed compared to a fixed 0.9 weighting in this case but offers greater adaptability across datasets and task complexities. A fixed weighting assumes constant loss importance, yet our analysis shows this ratio evolves dynamically. Using a fixed CL weight of 0.9 (and 0.1 for CE) may be suboptimal for complex tasks, as CE loss remains uncertain longer while CL loss stabilizes earlier.

**Confusion Matrices.** Figure 5 illustrates the confusion matrices subsequent to the final task to all test classes within the CIFAR, ImageNet, and GNSS datasets, employing MEMO (Zhou et al., 2023b) and our

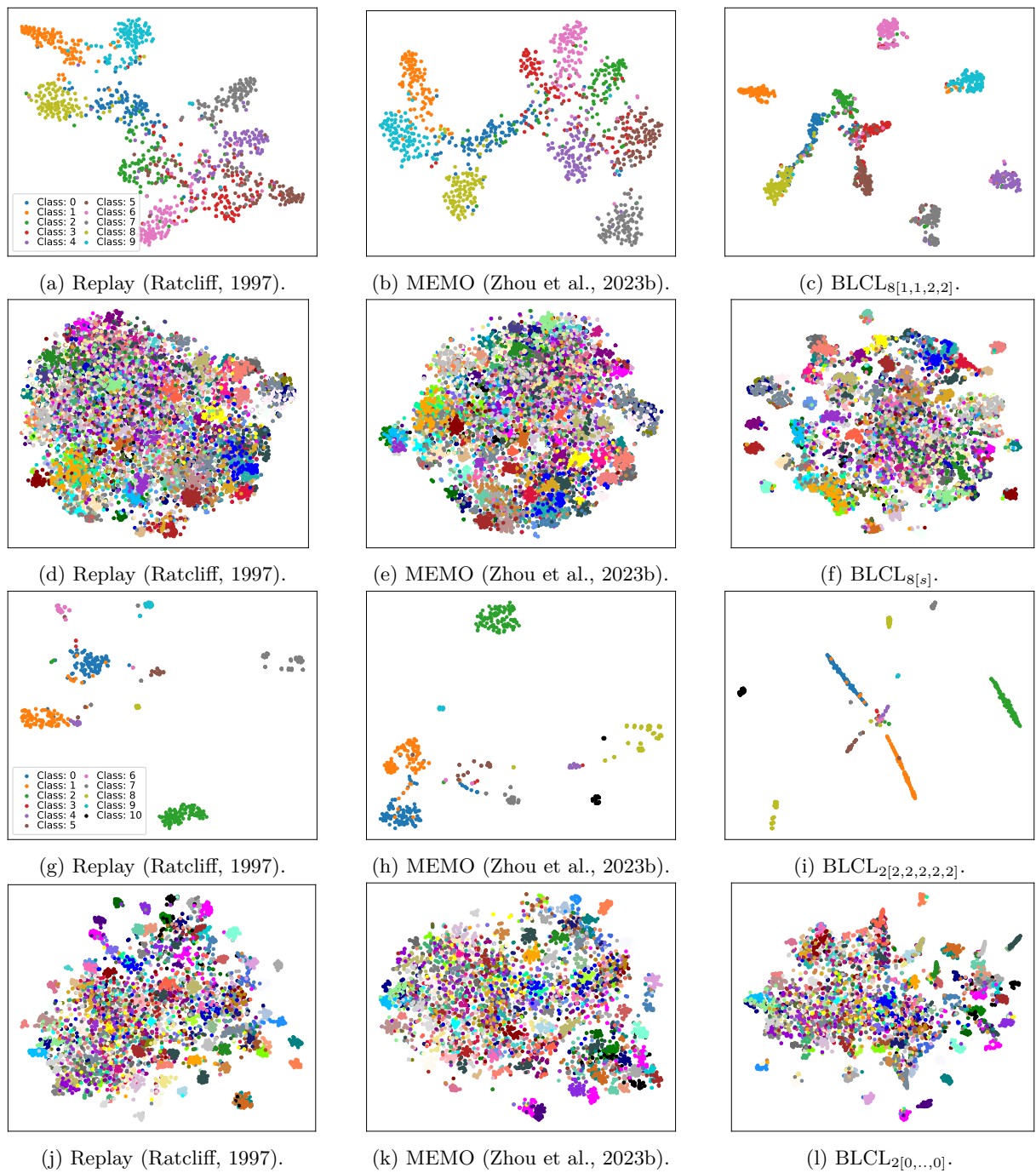

Figure 6: t-SNE plots for the CIFAR-10 (a to c), CIFAR-100 (d to f), GNSS (g to i), and ImageNet100 (j to k) datasets.

proposed technique denoted as $BLCL_{8[1,1,2,2]}$ and $BLCL_{2[2,2,2,2,2]}$ with Bayesian weighting, respectively. See the appendix, for confusion matrices on all methods. On the CIFAR-10 dataset, MEMO demonstrates significant performance superiority over Replay for specific classes 0, 1, 4, 5, 6, and 7 (refer to Figure 13e compared to Figure 5a). Notably, Replay exhibits pronounced confusion between classes 8 and 9 as well as the remaining classes, a misclassification mitigated by MEMO. Additionally, BLCL achieves a higher true positive rate than MEMO for classes 0, 2, 3, 4, and 9, thereby attaining an average accuracy of 86.57%, surpassing

MEMO's 84.78% (see Figure 5e). Our approach effectively mitigates misclassifications, particularly between classes 3 and 6. On the GNSS dataset, a distinct resemblance exists between classes 1 and other classes (refer to Figure 3), resulting in misclassifications. While MEMO exhibits a higher false positive rate compared to Replay, their false negative rates remain equal, leading to Replay's superior average accuracy of 95.45% versus MEMO's 95.34% (compare Figure 16e with Figure 5c). Leveraging enhanced representation learning via contrastive loss, our BLCL method effectively separates positive classes (3 to 10) into distinct clusters, thus reducing the false positive rate and yielding a higher F2-score of 0.678 while simultaneously decreasing the false negative rate (refer to Figure 5g).

**Cluster Analysis.** Subsequently, we compare the Davies-Bouldin (DB) (Davies & Bouldin, 1979) and Caliński-Harbasz (CH) (Caliński & Harabasz, 1972) scores. DB is defined as the mean similarity measure between each cluster and its most similar cluster. This measure of similarity is the ratio of within-cluster distances to between-cluster distances. Consequently, clusters exhibiting greater separation and lesser dispersion will yield a superior score (Halkidi et al., 2001; Ros et al., 2023; Thrun, 2011). CH is a variance-ratio criterion, where a higher CH score relates to a model with better defined clusters, and is defined as the ratio of the sum of inter-clusters dispersion and of intra-cluster dispersion for all clusters. Table 1 to Table 3 summarizes the scores attributed to Finetune, EWC, LwF, iCarl, Replay, MEMO, DER, FOSTER, and BLCL. Across all three datasets, BLCL demonstrates a lower DB score and higher CH score in comparison to the baseline methods. This observation supports the assertion that BLCL results in a representation characterized by lower inter-class distances and enhanced clustering efficacy.

**Embedding Analysis.** The feature embedding of the last convolutional layer, with an output size of 512, is depicted in Figure 6. This embedding is generated using t-distributed stochastic neighbor embedding (t-SNE) (van der Maaten & Hinton, 2008) with a perplexity of 30, an initial momentum of 0.5, and a final momentum of 0.8. On the CIFAR-10 dataset, the Replay method (Figure 6a) and MEMO method (Figure 6b) demonstrate poor separation of clusters, resulting in the overlap of classes 2, 3, and 4, thereby leading to misclassification. This observation aligns with the findings presented earlier using confusion matrices (refer to Figure 5a and 13e). Our proposed BLCL method (Figure 6c), which incorporates contrastive loss, significantly reduces the intra-class distance and increases inter-class distance. This is also evident for the CIFAR-100 dataset (refer to Figure 6d to 6f). However, some overlap persists, particularly for classes 0, 1, and 3, resulting in a higher rate of misclassification. Similar trends are observed in the GNSS dataset, where both Replay and MEMO methods exhibit overlapping samples with class 0 (Figure 6g and 6h), while the BLCL method only forms a small cluster with specific samples in the center of the embedding (Figure 6i).

# 6 Conclusion

In the context of the continual learning task, we proposed a method characterized by a dynamic specialized component. This component combines the CE loss with a prototypical contrastive loss by utilizing Bayesian learning principles. By employing dynamic loss weighting, our method effectively achieves a feature representation with reduced intra-class disparity and higher inter-class distance. Evaluations conducted on the CIFAR-10, CIFAR-100, ImageNet100, and GNSS datasets demonstrate that our proposed technique outperforms the state-of-the-art Replay, MEMO, and DER. Specifically, our method achieves accuracy rates of up to 86.57% on CIFAR-10, 70.43% on CIFAR-100, 79.73% on the ImageNet100, and 96.16% on the GNSS dataset with improved Davies-Bouldin and Caliński-Harabasz scores.

**Acknowledgments**

This work has been carried out within the DARCII project, funding code 50NA2401, supported by the German Federal Ministry for Economic Affairs and Climate Action (BMWK), managed by the German Space Agency at DLR and assisted by the Bundesnetzagentur (BNetzA) and the Federal Agency for Cartography and Geodesy (BKG).

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

# A    Appendix

## A.1    Data Augmentation

Due to the potential for highly unbalanced training dataset, such as the interfered classes within the GNSS dataset, we address this issue by balancing each class through data augmentation techniques (Fairstein et al., 2024; Qin et al., 2023; Treder-Tschechlov et al., 2023). Specifically, we augment the data to ensure a uniform distribution of 1,000 samples per class. We employ techniques provided by *torchvision.transforms*, including color jittering with a brightness value of 0.5 and a hue value of 0.3, Gaussian blur with a kernel size of $(5, 9)$ and sigma values of $(0.1, 5.0)$, random adjustments of sharpness with a sharpness factor of 2, and random horizontal and vertical flipping with a probability of 0.4. Values are chosen after a hyperparameter search.

## A.2    Selection of the Backbone Model

To identify the optimal backbone architecture for the generalized blocks, we trained 110 vision encoder models from Hugging Face (Wightman, 2019) on the complete GNSS classification dataset. Figure 7 provides a summary of the results across five different metrics: all classes, only positive interference classes, the binary score, the F2-score, and the F5-score. While various model sizes were evaluated, only the primary model names are visualized for clarity. Given the highly imbalanced nature of the dataset, all models yielded similar evaluation results for both the overall class evaluation and the binary metric. However, the ResNet18 model (visualized gray) outperformed all other models on the interference classes, prompting us to select ResNet as the backbone model.

## A.3    Dynamic Adjustment of Layer Structures

In deep neural networks, the deeper layers are primarily responsible for learning task-specific features, as they capture high-level abstractions crucial for distinguishing between different classes. ResNet comprises four specialized ResNet blocks, each containing four convolutional layers, as illustrated in Figure 2. In our approach, we dynamically modify only the final ResNet block, as it is the most specialized for task-specific feature extraction. To determine the appropriate depth for a new task, we compute its class similarity to previously learned tasks. If the new classes are similar to the previously learned ones, we assume that the earlier layers have already captured the essential features, thereby allowing us to reduce the number of active convolutional layers in the last block. We progressively remove layers one by one until the accuracy begins to decline, effectively performing a slight hyperparameter search to identify the maximum number of layers that can be removed without compromising performance. Conversely, for dissimilar classes, we retain all layers to ensure the model can learn new and distinct features. This adaptive adjustment facilitates efficient knowledge reuse while preserving the model's flexibility to accommodate new tasks. This approach was demonstrated to be effective for CIFAR-10 and GNSS, where two classes were introduced per task, allowing for gradual layer adjustment. However, in CIFAR-100, where ten classes were introduced per task, it proved more beneficial to retain all layers to maintain optimal accuracy.

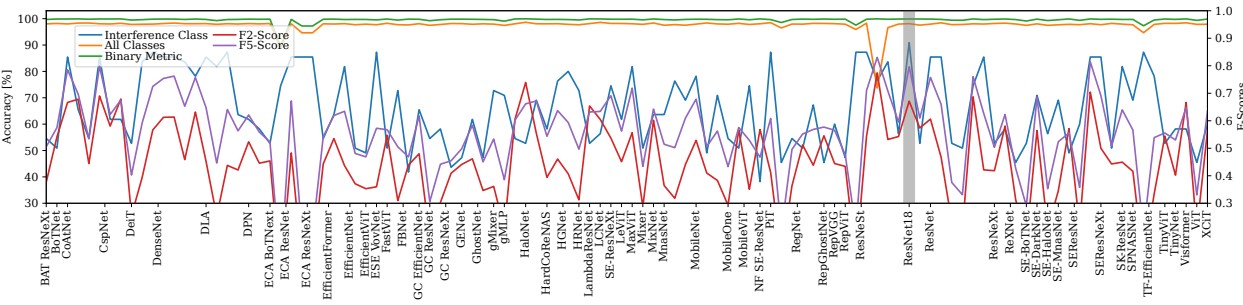

Figure 7: An evaluation of 110 vision encoder backbone models on the GNSS dataset has been conducted. The top-performing model is highlighted in gray.

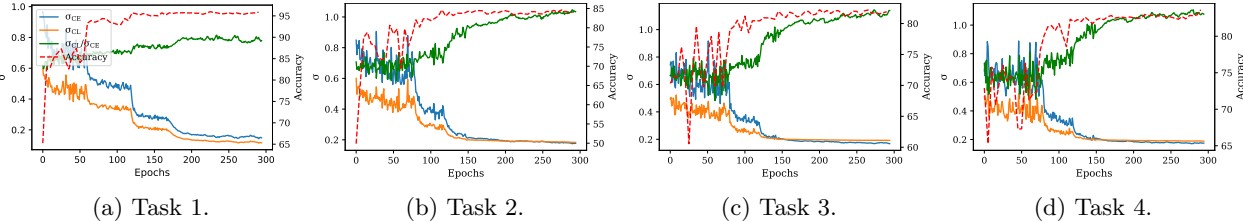

(a) Task 1.      (b) Task 2.      (c) Task 3.      (d) Task 4.

Figure 8: Analysis of $\sigma_1$, $\sigma_2$ and the classification accuracy for each task for the CIFAR-10 dataset.

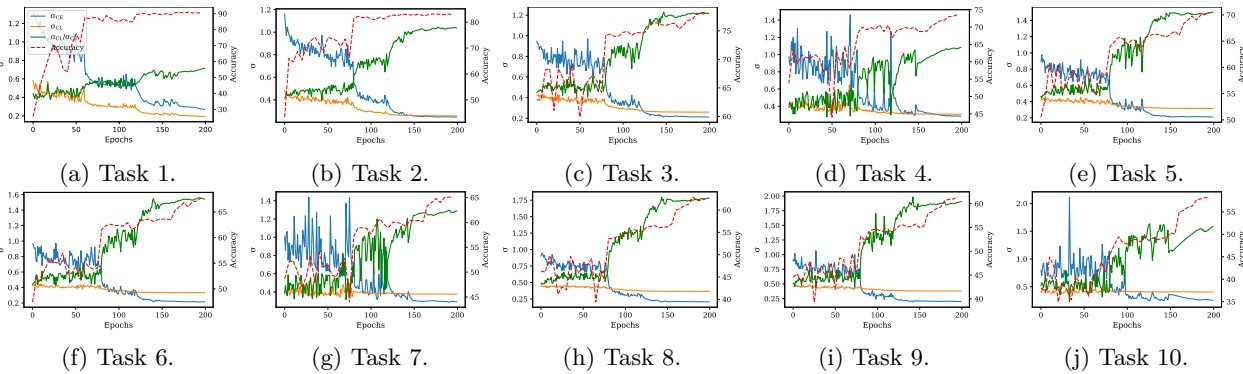

(a) Task 1.    (b) Task 2.    (c) Task 3.    (d) Task 4.    (e) Task 5.

(f) Task 6.    (g) Task 7.    (h) Task 8.    (i) Task 9.    (j) Task 10.

Figure 9: Analysis of $\sigma_1$, $\sigma_2$ and the classification accuracy for each task for the CIFAR-100 dataset.

## A.4 Bayesian Learning-driven Weighting

We next present an ablation study on the Bayesian learning-driven weighting of the cross-entropy (CE) and contrastive learning (CL) loss functions. Balancing CE and CL loss functions is essential because they capture complementary aspects of the learning process, ensuring a more comprehensive optimization strategy. CE focuses on maintaining class-specific discrimination by minimizing prediction errors within predefined categories, which helps the model align its outputs with known labels. On the other hand, CL encourages general feature extraction by promoting meaningful representations across samples regardless of explicit labels. If only the CL loss is optimized, the model may prioritize similarity relationships without anchoring them to categorical class distinctions, potentially leading to suboptimal decision boundaries and degraded performance on classification tasks. By simultaneously applying both losses, the model benefits from both structured learning (CE) and representation enhancement (CL). Dynamic weighting becomes necessary because the importance of each objective may shift throughout the training process. Early in training, stronger emphasis on CL may help develop better feature representations, while CE becomes more important as decision boundaries are refined. Therefore, a dynamic balance between the two losses ensures adaptive learning, improving generalization and robustness in the final model. The CE loss quantifies performance on classification tasks, with a higher weight indicating that classification accuracy is either of greater importance or subject to higher uncertainty. Conversely, the CL loss facilitates the learning of meaningful feature representations by contrasting positive and negative pairs, where a higher weight signifies a focus on developing robust embeddings. The posterior distributions of the weights assigned to each loss function provide key insights. Broad posterior distributions indicate significant uncertainty regarding the relative importance of the losses. Furthermore, shifts in the weight distributions during training may reflect changing task priorities — for instance, emphasizing feature representation learning in the early stages and classification accuracy in later stages. When the CE loss diminishes and the model demonstrates confidence in classifications, the weighting of the CE loss may decrease, reallocating focus to the CL loss. Similarly, if the CL loss plateaus, its weighting may reduce to prioritize fine-tuning classification performance.

Figure 8, Figure 9, Figure 10, and Figure 11 illustrate the dynamic weighting mechanism applied to the CE and CL loss functions throughout training for all tasks on the CIFAR-10, CIFAR-100, ImageNet100, and GNSS datasets, respectively. As $\sigma_1$ and $\sigma_2$, the noise parameters associated with the variables $\mathbf{y}_1$ and

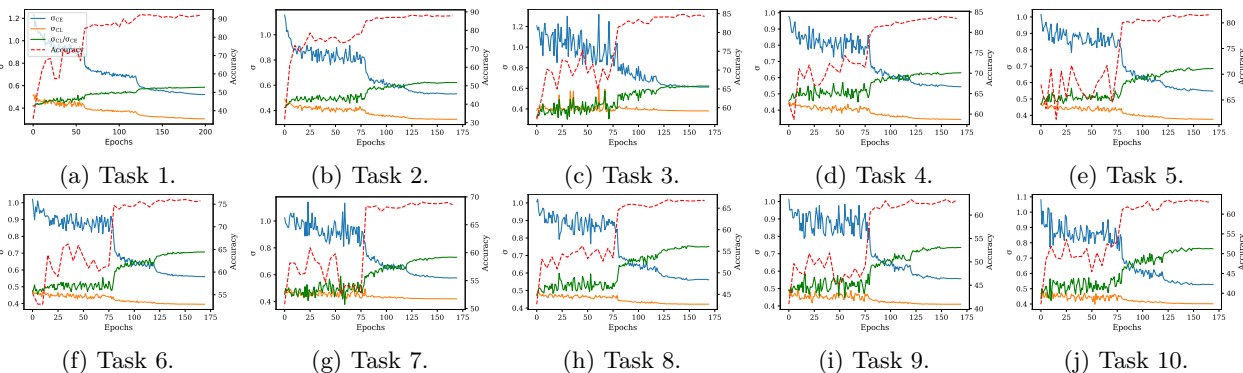

Figure 10: Analysis of $\sigma_1$, $\sigma_2$ and the classification accuracy for each task for the ImageNet100 dataset.

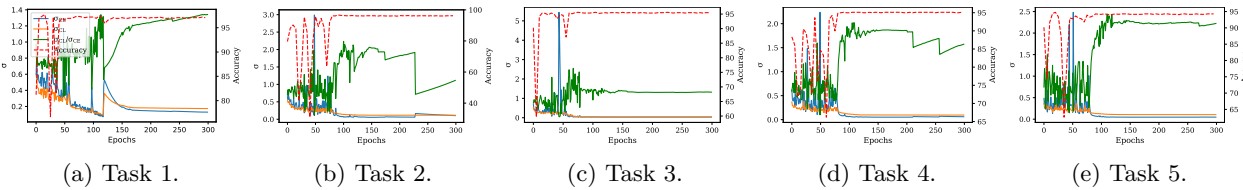

Figure 11: Analysis of $\sigma_1$, $\sigma_2$ and the classification accuracy for each task for the GNSS dataset.

$\mathbf{y}_2$, respectively (introduced in Section 3.3), increase, the weights assigned to $\mathcal{L}_{CE}$ and $\mathcal{L}_{CL}$ correspondingly decrease. This adjustment acts as a regularization mechanism for the noise terms. This phenomenon is particularly evident during the early stages of each task, where the introduction of new classes increases noise levels, leading to a reduction in the weighting of both loss functions. Early in training, the $\sigma$ value for the CE loss is high, meaning that its weighting is relatively low. In contrast, the $\sigma$ value for the CL loss is lower, indicating a higher weighting. This suggests that, initially, the model prioritizes learning meaningful representations through contrastive learning rather than optimizing for classification accuracy. As training progresses, the $\sigma$ for CE loss decreases while the $\sigma$ for CL loss increases, leading to a shift where the model places more emphasis on classification accuracy. The initial fluctuation in the weights, particularly for the CE loss, may signify the Bayesian framework's attempt to model and adapt to the high uncertainty in the early stages of training. Over time, the smoother trends in the weights imply that the uncertainty has diminished as the model gains confidence in its predictions and representations. The decrease in weighting for both loss functions over epochs acts as an implicit regularization mechanism. This ensures that the model does not overfit to specific aspects of the loss functions and instead balances classification performance and representation learning. By around 200, 150, and 200 epochs, respectively, both weights stabilize, indicating that the contributions of the CE and CL losses have reached an equilibrium. This suggests that the model has adequately learned both the task-specific and representation-related objectives.

### A.5 Hardware & Training Setup

All experiments are conducted utilizing Nvidia Tesla V100-SXM2 GPUs with 32 GB VRAM, equipped with Core Xeon CPUs and 192 GB RAM. We use the vanilla Adam optimizer with a learning rate set to 0.1, a decay rate of 0.1, a batch size of 128, and train each task for 300 epochs.

After each task we reduce the number of exemplars of previous classes. For example, the CIFAR-100 dataset has 10 tasks with each 10 classes. For the first task, the dataset contains 500 samples for each class for 10 classes. For the second task, the dataset contains 200 samples for each of the previous 10 classes and 500 samples for the new 10 classes. For the third task, the dataset contains 100 for each of the previous 20 classes and 500 samples for the new 10 classes. For the fourth task, the dataset contains 66 samples for each of the previous 30 classes and 500 samples for the new 10 classes. In this way, the number of samples of the previous classes decrease to accommodate samples of new classes in a fixed memory.

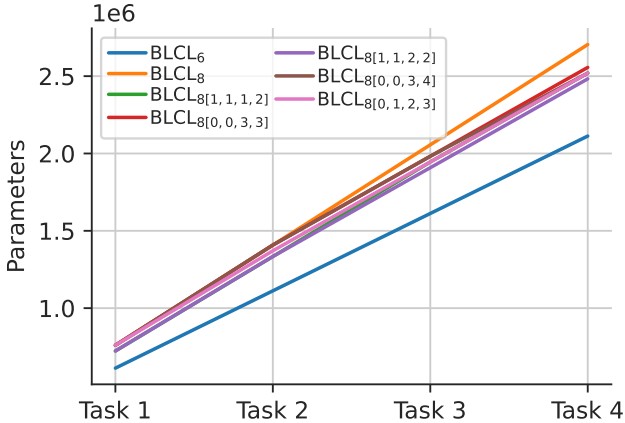

Figure 12: Number of model parameters for each task.

Table 5: Model properties.

| Dataset | Method | Backbone | Model Size | Samples | Sample Size | Total |
|---|---|---|---|---|---|---|
| **CIFAR-10 and CIFAR-100** | Replay (Ratcliff, 1997) | ResNet32 | 1.75 MB | 7,431 | 21.76 MB | 23.51 MB |
| | iCarl (Rebuffi et al., 2017) | ResNet32 | 1.75 MB | 7,431 | 21.76 MB | 23.51 MB |
| | DER (Yan et al., 2021) | ResNet32 | 17.68 MB | 2,000 | 5.85 MB | 23.53 MB |
| | MEMO (Zhou et al., 2023b) | ResNet32 | 13.83 MB | 3,312 | 9.70 MB | 23.53 MB |
| | BLCL (ours) | ResNet50 | 15.85 MB | 2,000 | 5.85 MB | **20.85** MB |
| **GNSS** | Replay (Ratcliff, 1997) | ResNet18 | 42.60 MB | 4,970 | 713.50 MB | 756.10 MB |
| | iCarl (Rebuffi et al., 2017) | ResNet18 | 42.60 MB | 4,970 | 713.50 MB | 756.10 MB |
| | DER (Yan et al., 2021) | ResNet18 | 468.00 MB | 2,000 | 287.00 MB | 755.00 MB |
| | MEMO (Zhou et al., 2023b) | ResNet18 | 362.80 MB | 2,739 | 393.20 MB | 755.00 MB |
| | BLCL (ours) | ResNet18 | 362.80 MB | 2,000 | 287.00 MB | **649.80** MB |
| **ImageNet100** | Replay (Ratcliff, 1997) | ResNet18 | 42.60 MB | 4,970 | 713.50 MB | 756.10 MB |
| | iCarl (Rebuffi et al., 2017) | ResNet18 | 42.60 MB | 4,970 | 713.50 MB | 756.10 MB |
| | DER (Yan et al., 2021) | ResNet18 | 468.00 MB | 2,000 | 287.00 MB | 755.00 MB |
| | MEMO (Zhou et al., 2023b) | ResNet18 | 362.80 MB | 2,739 | 393.20 MB | 755.00 MB |
| | BLCL (ours) | ResNet18 | 362.80 MB | 2,739 | 393.20 MB | 755.00 MB |

## A.6 Parameter Counts

Figure 12 presents a detailed overview of the parameter counts for various configurations of the BLCL architecture, including $BLCL_6$, $BLCL_8$, and specialized configurations $BLCL_{8[s]}$, across all tasks within the CIFAR-10 dataset. The configurations differ based on the number and arrangement of dynamic specialized components. Transitioning from $BLCL_6$ to $BLCL_8$, which increases the number of blocks in the specialized component from 6 to 8, leads to a rise in parameter count from approximately $2.1 \cdot 10^6$ to $2.7 \cdot 10^6$. This increase in model complexity correlates with an improvement in accuracy (as reported in Table 2), highlighting the trade-off between model size and performance. The configurations $BLCL_8$ feature selective reductions in the number of specialized blocks for different tasks, as indicated in the legend. For example, $BLCL_8[0, 0, 0, 0]$ represents the smallest architecture, with minimal or no specialized components. Despite the reduced parameter count, these configurations maintain competitive accuracy, showcasing the efficiency of dynamic specialization in balancing model size and performance. As observed in Figure 12, the parameter count increases progressively with each additional task, reflecting the adaptive inclusion of specialized components. Configurations with more specialized blocks, such as $BLCL_8[1, 1, 1, 2]$ and $BLCL_8[1, 1, 2, 3]$, experience a steeper increase in parameter count compared to $BLCL_6$. This scaling behavior demonstrates the flexibility of the architecture to adapt to task complexity while controlling overall model growth. The comparison of $BLCL_8[1, 1, 2, 3]$ with simpler configurations like $BLCL_8[0, 0, 0, 0]$ or $BLCL_8[0, 0, 3, 3]$ illustrates how selective reduction of parameters can retain performance. This observation suggests that the dynamic allocation of specialized components effectively balances the need for task-specific learning and model generalization. Table 5 contextualizes these configurations by comparing their parameter counts, model sizes,

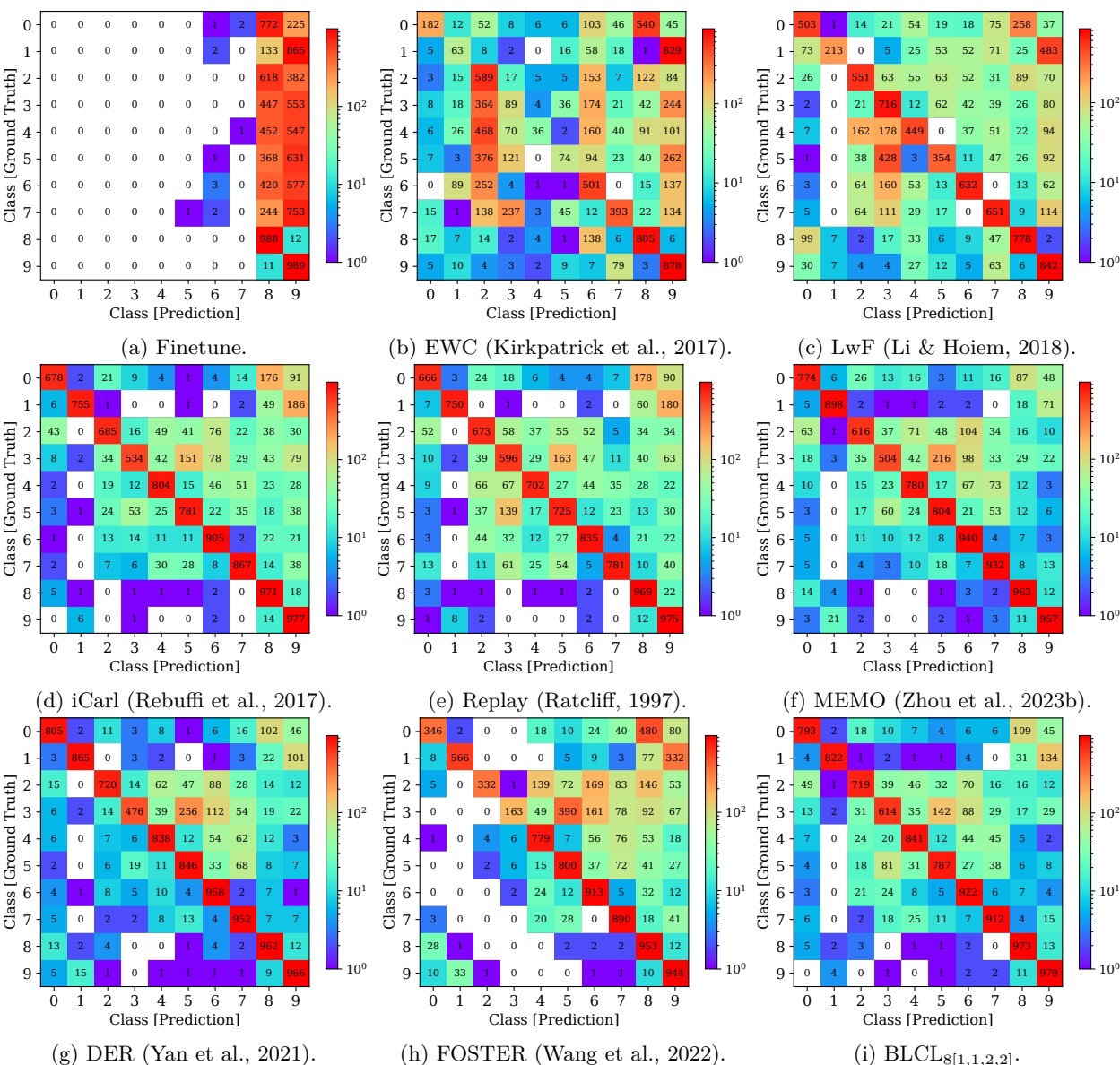

Figure 13: Confusion matrices of state-of-the-art methods and our Bayesian BLCL approach for the CIFAR-10 dataset.

and memory footprints with baseline methods such as Replay, iCarl, and DER. The BLCL configurations achieve competitive or superior performance while maintaining manageable model sizes.

## A.7 Embedding Analysis After Each Task

Figure 13, Figure 14, Figure 15, and Figure 16 present the confusion matrices for all methods on the CIFAR-10, CIFAR-100, and GNSS datasets, respectively. Figure 17, Figure 18, Figure 19, and Figure 20 present the t-SNE visualizations for Replay, MEMO, DER, FOSTER, and BLCL on these three datasets, respectively.

Figure 21 visualizes the feature embeddings after each of the tasks. Specifically for the CIFAR-10 dataset (a to d), when adding clusters from new tasks, a large inter-class distance between clusters from previous classes are maintained. However, for the GNSS dataset (e to i), the samples from classes 3 and 4 from task 2 are added in-between the classes 0 to 2 due to a high similarity between the classes.

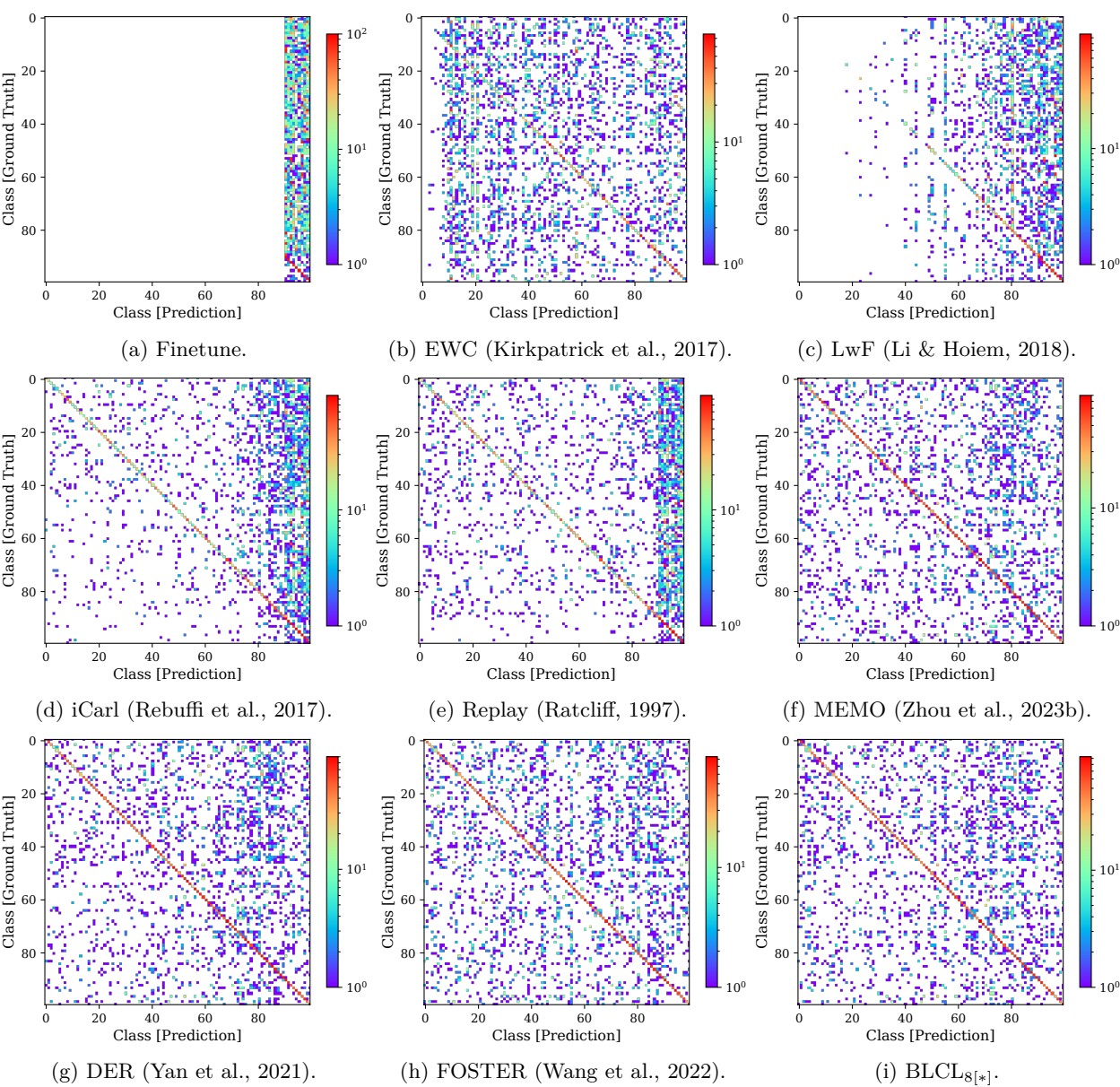

Figure 14: Confusion matrices of state-of-the-art methods and our Bayesian BLCL approach for the CIFAR-100 dataset.

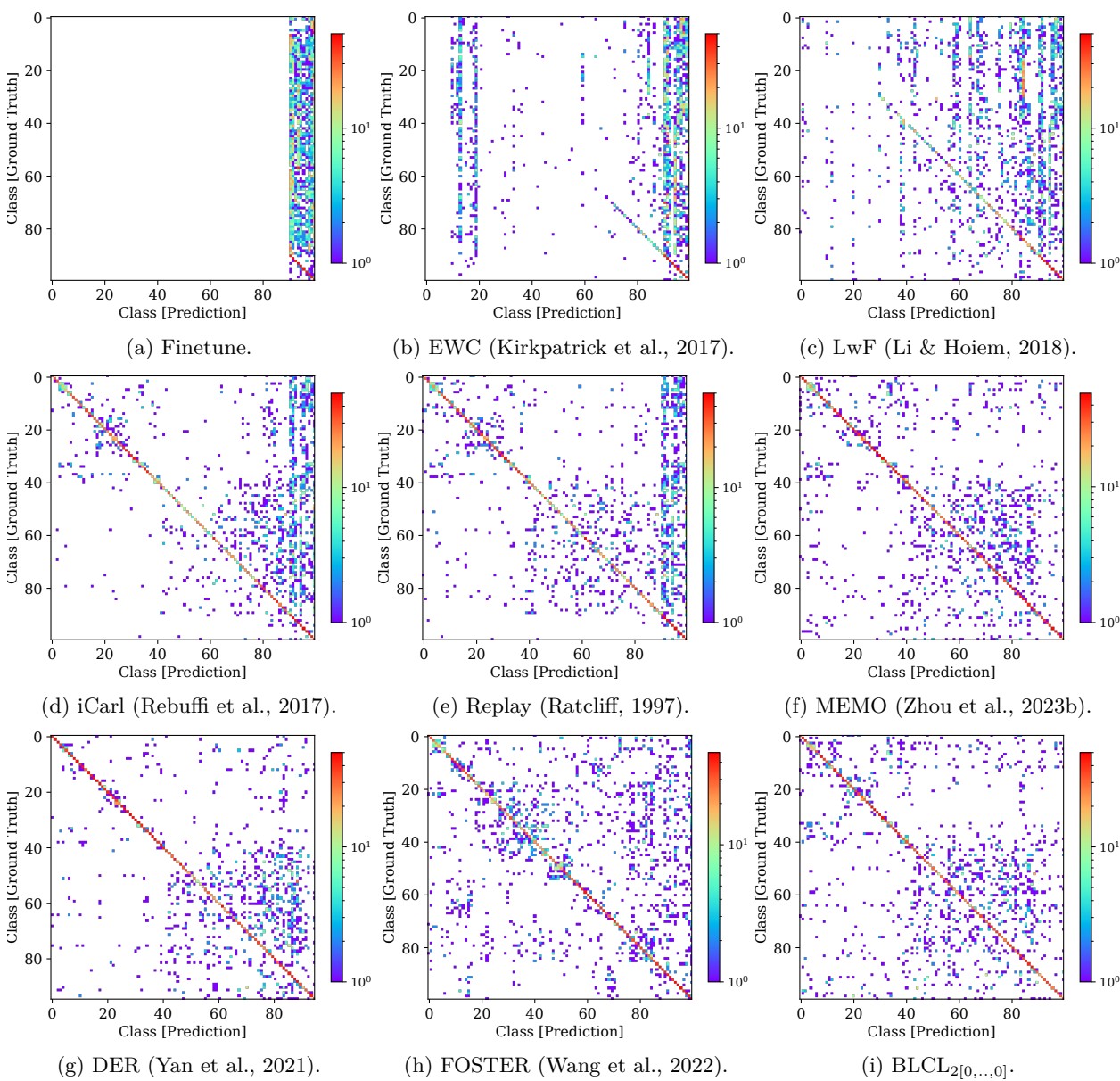

Figure 15: Confusion matrices of state-of-the-art methods and our Bayesian BLCL approach for the ImageNet100 dataset.

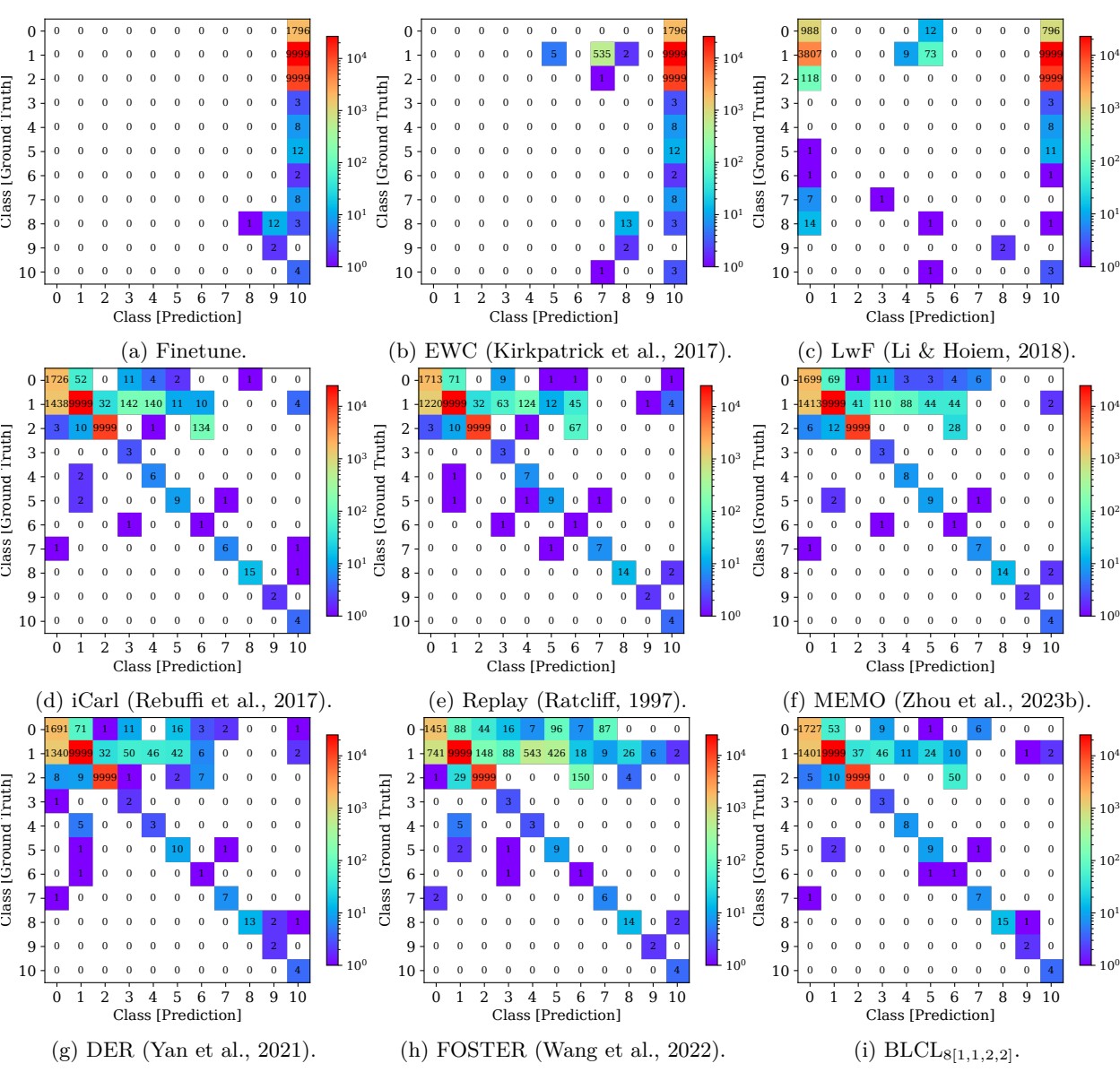

Figure 16: Confusion matrices of state-of-the-art methods and our Bayesian BLCL approach for the GNSS dataset.

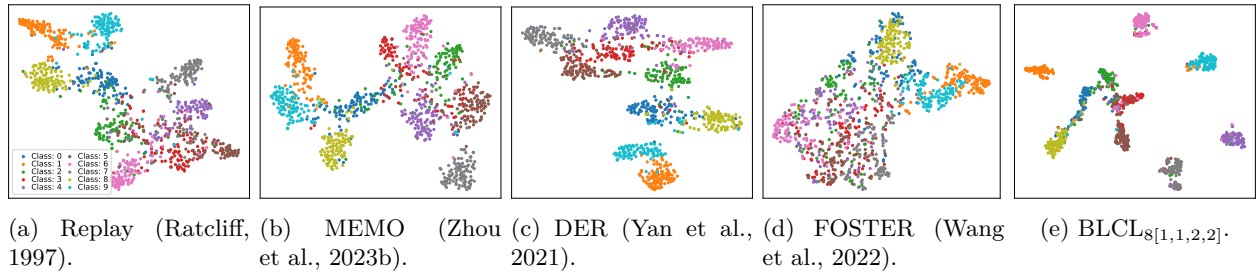

(a) Replay (Ratcliff, 1997).  (b) MEMO (Zhou et al., 2023b).  (c) DER (Yan et al., 2021).  (d) FOSTER (Wang et al., 2022).  (e) $\text{BLCL}_{8[1,1,2,2]}$.

Figure 17: t-SNE (van der Maaten & Hinton, 2008) plots for the CIFAR-10 dataset.

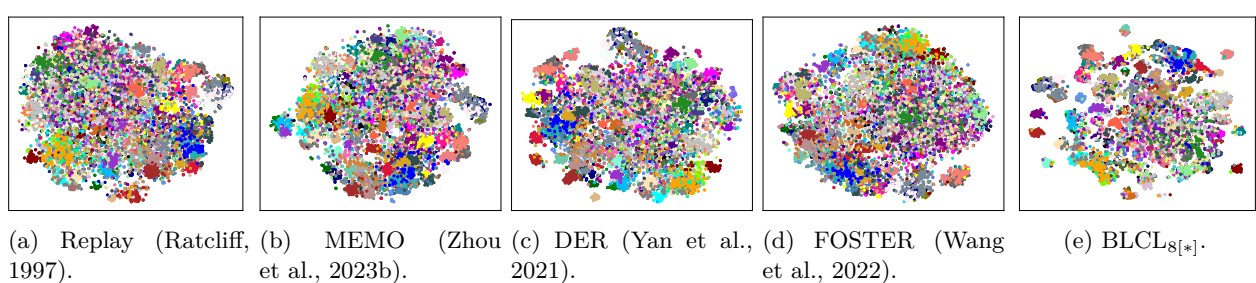

(a) Replay (Ratcliff, 1997).  (b) MEMO (Zhou et al., 2023b).  (c) DER (Yan et al., 2021).  (d) FOSTER (Wang et al., 2022).  (e) $\text{BLCL}_{8[*]}$.

Figure 18: t-SNE (van der Maaten & Hinton, 2008) plots for the CIFAR-100 dataset.

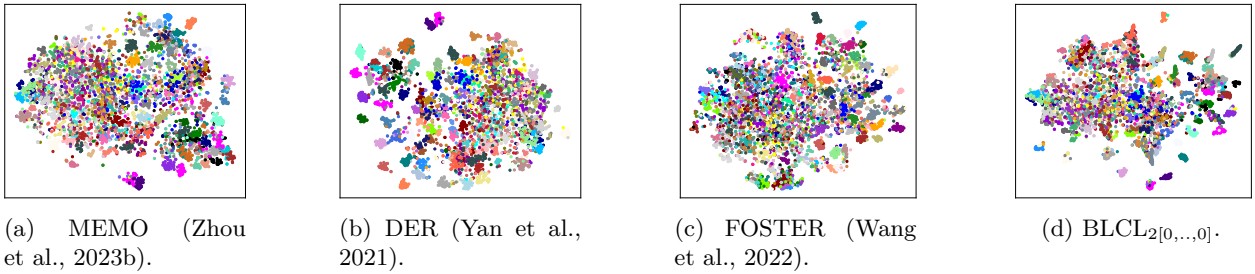

(a) MEMO (Zhou et al., 2023b).  (b) DER (Yan et al., 2021).  (c) FOSTER (Wang et al., 2022).  (d) $\text{BLCL}_{2[0,..,0]}$.

Figure 19: t-SNE (van der Maaten & Hinton, 2008) plots for the ImageNet100 dataset.

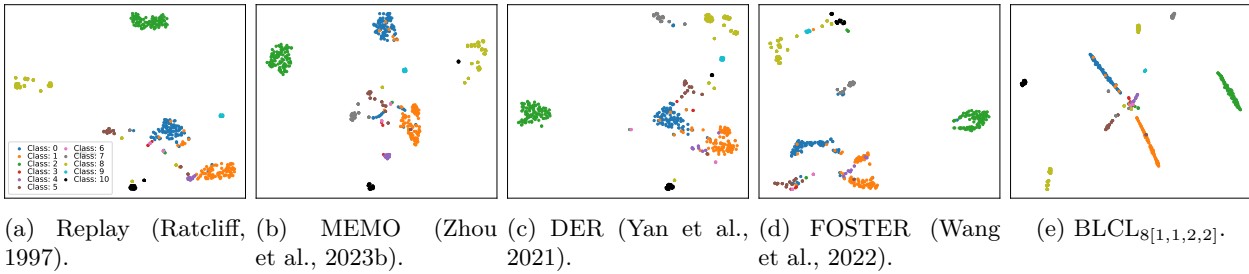

(a) Replay (Ratcliff, 1997).  (b) MEMO (Zhou et al., 2023b).  (c) DER (Yan et al., 2021).  (d) FOSTER (Wang et al., 2022).  (e) $\text{BLCL}_{8[1,1,2,2]}$.

Figure 20: t-SNE (van der Maaten & Hinton, 2008) plots for the GNSS dataset.

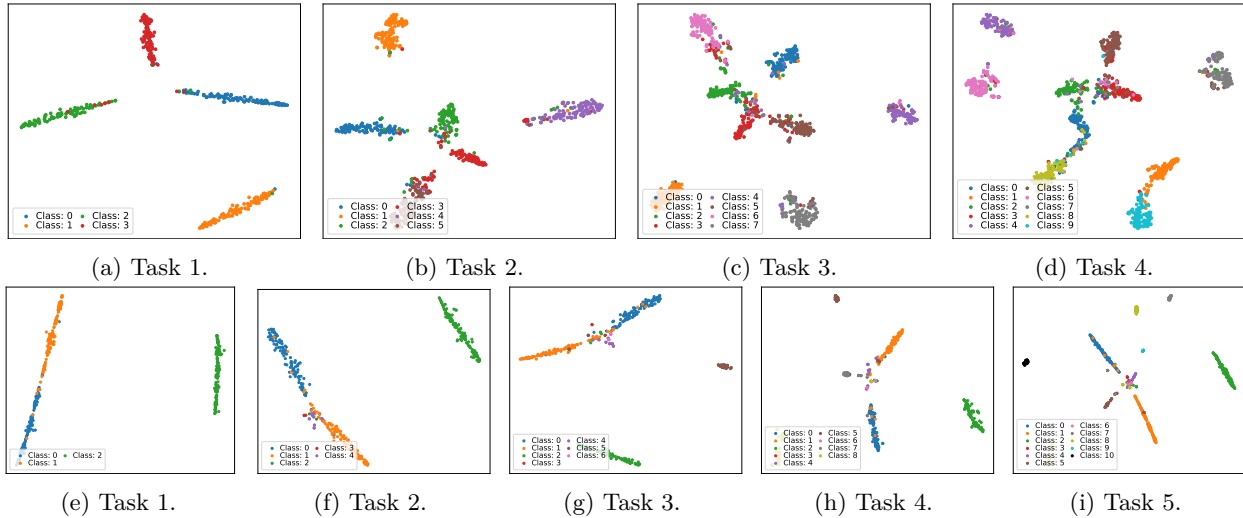

(a) Task 1.     (b) Task 2.     (c) Task 3.     (d) Task 4.

(e) Task 1.     (f) Task 2.     (g) Task 3.     (h) Task 4.     (i) Task 5.

Figure 21: t-SNE (van der Maaten & Hinton, 2008) plots for the CIFAR-10 (a to d) and GNSS (e to i) datasets after each task for our BLCL method.

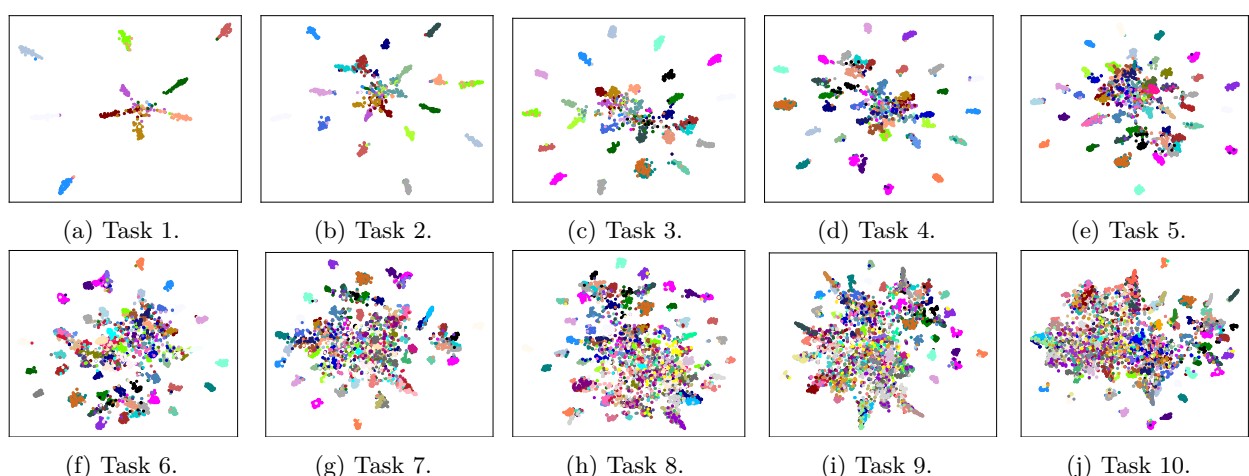

(a) Task 1.     (b) Task 2.     (c) Task 3.     (d) Task 4.     (e) Task 5.

(f) Task 6.     (g) Task 7.     (h) Task 8.     (i) Task 9.     (j) Task 10.

Figure 22: t-SNE (van der Maaten & Hinton, 2008) plots for the ImageNet100 dataset after each task for our BLCL method.

