# OpenReview forum: "Bayesian Learning-driven Prototypical Contrastive Loss for Class-Incremental Learning"
_TMLR — Accepted by TMLR_

### Review · Reviewer_og1a · 2025-01-31

**Summary Of Contributions:**

This paper introduces a Bayesian Learning-driven Prototypical Contrastive Loss (BLCL) for Class-Incremental Learning (CIL). The core idea is to dynamically balance cross-entropy (CE) loss and contrastive learning (CL) loss using homoscedastic task uncertainty. The method also averages the specialized component weights at the end of each task's learning process, thereby alleviating catastrophic forgetting in CIL. The approach is evaluated on CIFAR-10, CIFAR-100, and a GNSS-based dataset, demonstrating improved performance compared to state-of-the-art methods.

**Audience:**

Yes

**Claims And Evidence:**

No

**Requested Changes:**

1. Clarify the motivation for Bayesian-based loss weighting and provide stronger theoretical or empirical justification for its necessity in balancing CE and CL losses.
2. Revise and expand the formulation of Equation (5) to explicitly define all optimization variables and correct potential errors in the equation.
3. Include algorithm pseudocode for the proposed method to improve clarity and reproducibility.

**Strengths And Weaknesses:**

**Strengths**
1. The paper is well-structured and clearly written.
2. The proposed Bayesian-driven loss weighting strategy for balancing CE loss and CL loss appears promising for continual learning scenarios.
3. Extensive experiments on CIFAR-10, CIFAR-100, and a GNSS-based dataset validate the effectiveness of the approach.
4. The confusion matrix and clustering visualizations provide additional insights into the effectiveness of the proposed method.

**Weaknesses**
1. The motivation for using Bayesian learning in weighting loss functions for continual learning needs clearer justification. In Section 3.3, the paper states that CL loss can incrementally enhance class prototypes. If this is the case, why is it necessary to balance CE loss and CL loss rather than optimizing CL loss alone? If applying both losses simultaneously is essential and dynamic weighting provides better results, the paper lacks an in-depth analysis of the roles of CE loss and CL loss during training. Furthermore, there is no ablation study comparing the performance when using only CE loss or only CL loss.

2. Section 3.2 states: "To maintain adaptability to varying complexities of new classes across tasks, we dynamically adjust each task’s layer structure." However, the paper does not clearly explain how the layer structure is dynamically adjusted during continual learning.

3. It is unclear whether $\sigma_1$ and $\sigma_2$ in Equation (5) are trainable parameters or optimized variables with an analytical solution. The paper should provide a detailed explanation of all optimization variables in the formulation. Additionally, in the first line of Equation (5), the rightmost expression appears incorrect—based on Equation (2), it should involve addition rather than multiplication.

4. The paper lacks a pseudocode representation of the proposed method, which would help readers better understand the implementation details.

---

> ### Author Response · Authors · 2025-02-13
> **Answer to Reviewer og1a**
>
> We would like to thank the reviewer for the positive feedback and highlighting the strengths of our manuscript.
>
> We address the raised questions as follows:
>
> 1. In the current version of the paper, we have briefly introduced the weighting of the cross-entropy (CE) and contrastive learning (CL) loss functions, with a more detailed explanation provided in Appendix A.2. To improve the introduction, we will include a more thorough motivation for this in Section 3:
> Balancing CE and CL loss functions is essential because they capture complementary aspects of the learning process, ensuring a more comprehensive optimization strategy. Cross-entropy focuses on maintaining class-specific discrimination by minimizing prediction errors within predefined categories, which helps the model align its outputs with known labels. On the other hand, contrastive learning encourages general feature extraction by promoting meaningful representations across samples regardless of explicit labels. If only the contrastive learning loss is optimized, the model may prioritize similarity relationships without anchoring them to categorical class distinctions, potentially leading to suboptimal decision boundaries and degraded performance on classification tasks. By simultaneously applying both losses, the model benefits from both structured learning (CE) and representation enhancement (CL). Dynamic weighting becomes necessary because the importance of each objective may shift throughout the training process. Early in training, stronger emphasis on CL may help develop better feature representations, while CE becomes more important as decision boundaries are refined. Therefore, a dynamic balance between the two losses ensures adaptive learning, improving generalization and robustness in the final model.
>
> 2. In deep neural networks, the deeper layers are primarily responsible for learning task-specific features, as they capture high-level abstractions crucial for distinguishing between different classes. ResNet comprises four specialized ResNet blocks, each containing four convolutional layers, as illustrated in Figure 2. In our approach, we dynamically modify only the final ResNet block, as it is the most specialized for task-specific feature extraction. To determine the appropriate depth for a new task, we compute its class similarity to previously learned tasks. If the new classes are similar to the previously learned ones, we assume that the earlier layers have already captured the essential features, thereby allowing us to reduce the number of active convolution layers in the last block. We progressively remove layers one by one until the accuracy begins to decline, effectively performing a slight hyperparameter search to identify the maximum number of layers that can be removed without compromising performance. Conversely, for dissimilar classes, we retain all layers to ensure the model can learn new and distinct features. This adaptive adjustment facilitates efficient knowledge reuse while preserving the model’s flexibility to accommodate new tasks. This approach was demonstrated to be effective for CIFAR-10 and GNSS, where two classes were introduced per task, allowing for gradual layer adjustment. However, in CIFAR-100, where ten classes were introduced per task, it proved more beneficial to retain all layers to maintain optimal accuracy. We will further clarify this in the updated version of the paper.
>
> 3. The hyperparameters $\sigma_1$ and $\sigma_2$ are trainable parameters, initially set to one, indicating equal weighting for both. The update process for $\sigma_1$ and $\sigma_2$ will be presented in the pseudocode, and a detailed explanation of all optimization variables will be included in the updated paper.
>
> 4. We appreciate the reviewer’s request for pseudocode and will include it in Section 3 of the updated paper as suggested.

---

> > ### Comment · Reviewer_og1a · 2025-02-21
> >
> > Thank you for the author's response and the content added to the paper, which has resolved my doubts about the dynamic adjustment strategy and the hyperparameters $\sigma_1$ and $\sigma_2$. However, I still have the following two concerns:
> > 1. Lack of ablation experiments using only the CE loss or only the CL loss.
> > 2. The algorithm pseudocode usually does not include the initialization in line 3. Please consider deleting it or correcting the typo in line 3. And, if pseudocode is similar to PyTorch code, please note it (PyTorch-like).

---

> > > ### Author Response · Authors · 2025-02-21
> > > **Experiments on CE/CL-only and Pseudocode**
> > >
> > > We sincerely appreciate the reviewer’s feedback on our modifications to the dynamic adjustment strategy and hyperparameters.
> > >
> > > 1. Additionally, we have now included ablation experiments using only the CE loss and only the CL loss functions. Further results have been added in Section 5.
> > > 2. Regarding the pseudocode, we have removed the initialization in line 3 and included a note indicating that it follows Python and PyTorch-like syntax.

---

> > > > ### Comment · Reviewer_og1a · 2025-02-24
> > > >
> > > > I appreciate the authors' revisions addressing my comments. The updated manuscript now adequately resolves all my concerns raised during evaluation.

---

### Review · Reviewer_ysVo · 2025-02-09

**Summary Of Contributions:**

This paper proposes to mitigate catastrophic forgetting in class-incremental learning by (1) a Bayesian learning technique that dynamically balances the contrastive loss and cross entropy loss; (2) leveraging the MEMO architecture that trains a new specialized block per new
task. Experiments on CIFAR-10, CIFAR-100, and GNSS data validate the effectiveness of the proposed method.

**Audience:**

Yes

**Claims And Evidence:**

Yes

**Requested Changes:**

Please address my previous concerns and  improve the presentation.

**Strengths And Weaknesses:**

### Strengths

1. In general, the idea makes sense and the experiments show slight better performance compared to previous approaches.
2. The analysis regarding to the clusters and embeddings is somehow informative.

### Weaknesses

1. The presentation should be improved, where clearer definitions can make the paper easier to understand.
    - For example, it's better to define explicitly $BLCL_l[s]$, where $BLCL_8[1,1,2,2]$ should also be clearly explained. If I understand correctly, this is related to the number of tasks, the number of blocks, and the five possible architectures depicted in Figure 2.
    - Some sentences are written in a quite strange way, for example, "As σ 1, respectively σ 2, the noise parameters associated with variable y 1, respectively y 2, increase, the weight assigned to L CE, respectively L CL, diminishes."
2. The novelty is limited, mainly based on MEMO by adding a Bayesian approach to balance the two losses.
3. In terms of the Bayesian approach itself, I have the following question:
    - Given the cross entropy loss and contrastive loss could be different in their scales, does it really work to directly use the variance to balance the importance between them? Did you introduce any additional hyper-parameter to control this?
4. "Evaluation of dynamic specialized components reveals that smaller blocks are preferable for the first task." could you analyze the reason? Does this hold for all datasets and any task order? I feel this may be related to the complexity of the task.
5. Will the authors release the code to promise reproducibility?

---

> ### Author Response · Authors · 2025-02-14
> **Answer to Reviewer ysVo**
>
> Thank you for taking the time to review our manuscript. We sincerely appreciate your valuable comments and suggestions, which have helped us improve our work. We have carefully addressed each point and believe that the revised version reflects your insightful feedback. After we included all changes in the manuscript, we will upload an updated version.
>
> 1)
> - We appreciate the reviewer's comment. As an example, we have specified the number of blocks and the configuration of the final layers for BLCL8[1,1,2,2].
>
> - We will refine the wording of these sentences to enhance clarity and improve readability for the reader.
>
> 2) We appreciate the reviewer's feedback regarding the novelty of our work. While our approach builds upon MEMO, our key contribution lies in the integration of a Bayesian framework to dynamically balance the two losses. Unlike MEMO, which relies on a fixed or heuristically chosen loss weighting strategy, our method introduces a principled probabilistic approach that adapts to varying levels of uncertainty in the data. These enhancements demonstrate that our method is not a trivial extension but rather a meaningful advancement that improves both performance and interpretability.
>
> 3)
> - Indeed, the CE loss and CL loss may have different scales, which motivates our use of a Bayesian weighting approach with trainable variance parameters $\sigma_1$ and $\sigma_2$. This approach is based on the formulation proposed by Kendall et al., which leverages Bayesian learning for multi-task deep learning to weight multiple loss functions by accounting for the homoscedastic uncertainty of each task. This enables the simultaneous learning of various quantities with different units or scales in both classification and regression settings. In this framework, the relative importance of different loss terms is dynamically adjusted based on their respective task-specific uncertainties.
>
> - In our experiments, we observed that the learned variance terms alone were sufficient to maintain a stable balance between the two losses. However, if necessary, an additional hyperparameter could be introduced to further regulate this balance.
>
> 4) The preference for smaller blocks in the first task is primarily attributed to the model’s lack of prior knowledge at the beginning of training. Since the first task serves as the foundation for feature extraction, a more compact model encourages the learning of generalizable representations while mitigating the risk of overfitting to specific patterns. However, this observation does not necessarily apply to all datasets and task orders. The complexity of the first task is a crucial factor. For smaller or simpler datasets, such as CIFAR-10 and GNSS, using smaller specialized blocks initially helps the model focus on learning generalizable features. In contrast, for more complex datasets, such as CIFAR-100, retaining larger blocks from the outset proves more beneficial. As the reviewer pointed out, this observation aligns with the intuition that the complexity of the first task influences the optimal network depth required at the start.
>
> 5) Thank you for your question! We have uploaded the source code in the supplementary zip file that accompanies the manuscript. Additionally, we are committed to ensuring reproducibility and will make the source code publicly available on GitHub for broader access.

---

> > ### Comment · Reviewer_ysVo · 2025-02-27
> >
> > Thanks for the authors' responses and the according revisions, which have resolved most of my concerns.

---

### Review · Reviewer_acqF · 2025-02-10

**Summary Of Contributions:**

The paper proposes a Bayesian approach to balance weighting between the cross-entropy loss (CE)  and constrastive loss (CL) for class-incremental continual learning.  Specifically, the paper explicitly models the uncertainty in the categorical distribution from CE and the Gaussian distribution from the CL as two hyper-parameters. Through algebraic operations, the two hyper-parameters emerge as weightings for the respective losses. The optimization objective is to jointly optimize the two losses and their uncertainty hyper-parameters, which also function as weightings for the losses.

Additionally, the paper also explores architectural variations for different datasets and tasks.

**Audience:**

Yes

**Broader Impact Concerns:**

The paper does not have obvious negative social impact.

**Claims And Evidence:**

Yes

**Requested Changes:**

* Could the authors make the interpretation of uncertainty and loss weights more explicit?

* Could the authors demonstrate the actual relative importance between the two losses such as a ratio between the weights as a plot in addition to fig.7?

**Strengths And Weaknesses:**

Strength:
- The paper explicitly models uncertainty in losses and factor them out as weightings. This is a novel application for class incremental continual learning.
- The paper performs experiments on real world datasets GNSS.


Weakness:

- **Interpretation of the uncertainty weighting** For both losses (CE and CL), the weighting is inversely proportional to the uncertainty parameters. In other words, higher weighting means lower uncertainty. This makes sense from the uncertainty perspective. However, this makes less sense from an optimization perspective. As shown in Fig.7, the weights continue to drop for both losses throughout training, this means that uncertainty rises during training. This interpretation seems self-contradicting.

- **Effectiveness of the uncertainty weighting** From Fig.7, we can observe that the weightings for both losses seem to have a very high correlation. When balancing losses, the relative importance of the two losses are more indicative of their contributions. However, the strong correlation between the weights seem to suggest that there is no actual adjustment of importance between the losses. Furthermore, from table 3, the $BLCL_{2[1,1,1,1,1]}$ Bayesian version underperformed the regular version with a fixed 0.9 weighting. This seems to suggest that the core contribution does not provide significant improvement.

---

> ### Author Response · Authors · 2025-02-14
> **Answer to Reviewer acqF**
>
> We would like to thank the reviewer for the constructive feedback, which has helped improve this manuscript. We would like to address the raised questions:
>
> 1) Thank you for pointing this out. We acknowledge that our figure represents variances ($\sigma_1$ and $\sigma_2$) and not the weighting itself ($1/\sigma_1$ and $1/\sigma_2$). Since the sigma values are decreasing, this indicates that the model is becoming more confident during training, leading to higher weighting of the corresponding losses. This aligns with the expected behavior of Bayesian uncertainty weighting: as training progresses and the model learns more reliable feature representations, the uncertainty reduces, and the loss terms receive greater influence in optimization. We will clarify this in the updated paper to ensure the correct interpretation of the plotted values.
>
> 2)
>
> - We acknowledge your concern regarding the correlation between the weightings of the two losses. To clarify this, we will add an additional plot visualizing the ratio ($\sigma_{CL}/\sigma_{CE}$) which explicitly represents the relative importance between the two losses. From our analysis, we observe that while the variances ($\sigma_{CE}$ and $\sigma_{CL}$) appear highly correlated in the first task, this stability does not necessarily hold across all tasks and datasets. Additionally, when we extend our analysis beyond the initial task and different datasets, we find that the ratio of the two variances changes more significantly. This suggests that the Bayesian uncertainty weighting does adjust dynamically rather than simply maintaining a fixed relative importance. In fact, as training progresses, we see increase in the $\sigma_{CL}/\sigma_{CE}$ ratio over training, indicating a shift in the model’s preference towards the CE loss. Given that the actual weight of each loss is inversely proportional to its variance ($1/\sigma$), this means the model gradually prioritizes CL loss over the CE loss as training continues and, in the end, prioritizes the CE loss after the CL loss is stabilized. Furthermore, this dynamic adaptation becomes more pronounced in later tasks and more complex datasets like CIFAR-100, where we observe greater shifts in the ratio across tasks. This confirms that Bayesian uncertainty weighting does not maintain a fixed balance but rather adjusts dynamically based on task complexity and training progression. To enhance clarity, we will incorporate these additional plots into the paper as requested.
>
> - Regarding the performance comparison in Table 3, while the Bayesian version underperformed relative to a fixed weighting of 0.9 in this specific setting, we believe its advantage lies in adaptability across different datasets and task complexities. A fixed weighting assumes that the relative importance of losses remains constant throughout training, but our analysis of $\sigma_{CL}/\sigma_{CE}$ shows that this ratio evolves dynamically. Using a fixed CL weight of 0.9 (implying a 0.1 weight for the CE loss) throughout training may not be optimal for more complex tasks since from our experiment we see that the CE loss remains uncertain for a longer period, whereas the CL loss stabilizes earlier. Our results show that constant weighting does not perform well for the CIFAR-10 and CIFAR-100 datasets, further highlighting the benefits of Bayesian uncertainty weighting in adapting to varying task complexities. We will further clarify this interpretation in the evaluation section.

---

### Review · Reviewer_NhZE · 2025-02-10

**Summary Of Contributions:**

The authors propose a CIL method BLCL which builds off of the MEMO architecture and includes Bayesian weighting to balance cross-entropy and contrastive learning losses. As shown in their visualizations, the method achieves feature representations with high inter-class distances. The authors show the method has a SOTA accuracy over recent methods on CIFAR10/100 and GNSS datasets.

**Audience:**

Yes

**Broader Impact Concerns:**

I do not see any ethical concerns for this paper.

**Claims And Evidence:**

No

**Requested Changes:**

R1 (related to W1): Could the authors report results on an additional dataset for image classification with higher resolution than CIFAR? in MEMO, the authors report results on ImageNet 1000.

R2 (related to W2): Can you please modify 3.2 to articulate which design choices are novel contributions and which come from MEMO or other? As Bayesian CL and contrastive loss in CL are not new concepts, it would be very helpful to help distinguish how your method is novel in a more concrete manner (my belief is that the method is novel, but stating the contributions more clearly will help the reader as the method contains many existing concepts).

R3 (related to W3): Could you please experiment on a more modern architecture as well? In MEMO, the authors report additional results using the vision transformer.

R4 (related to W4): could you include more details such as stdev so that the reader can judge the statistical significance of the SOTA result?

**Strengths And Weaknesses:**

Strengths:

S1: I think the authors have done a good job communicating the main ideas in clear Figures.

S2: The authors use a real-world use case with the GNSS Snapshot experiments. This greatly helps the paper in terms of impact by showing robustness beyond toy CIFAR experiments.

S3: I greatly appreciate the detailed analysis and visualizations in the experiments. Good job to the authors for this.

Weakness

W1: Class-incremental learning for image classification on small images (e.g., 32x32 CIFAR) with rigid task boundaries and labeled data has become a low-impact problem setting due to the saturation of results and lack of adaptability to real world settings. Including the GNSS experiments has greatly strengthened the paper. However, the paper could really use some results on higher-resolution images such as ImageNet (or something in between ImageNet and CIFAR) to demonstrate it is robust to modern ML benchmarks.

W2: In the method section, it would be great to clarify in section 3.2 which contributions are your own, and which part of the architecture description is from MEMO. It seems needed given that, in main contribution 1, you specific your method is similar to MEMO, and then in 3.2 you say you leverage MEMO.

W3: The only architectures explored are ResNet models, which are from 2016. Could the authors explore more modern architectures, such as the Vision transformer?

W4: the gains are quite slim in Tables 2 and 3. Without stdev reported, it is hard to tell whether this is statistically significant.

---

> ### Author Response · Authors · 2025-02-17
> **Answer to Reviewer NhZE**
>
> We would like to sincerely thank the reviewer for taking the time to review our paper. The constructive feedback and insightful comments have been incredibly helpful in improving the quality of our work. We have carefully considered the reviewer’s comments and made revisions accordingly, which we believe have strengthened the paper.
>
> 1. We thank the reviewer for recognizing the importance of the GNSS dataset as an evaluation dataset for real-world applications and for suggesting additional experiments on the ImageNet dataset. We have initiated further experiments on ImageNet and will include the results in the paper this week.
>
> 2. We have clarified the contributions of BLCL in relation to MEMO, which serves as the baseline method.
>
> 3. To identify the optimal backbone architecture for the generalized blocks, we trained 110 modern vision encoder models from Hugging Face on the complete GNSS classification dataset. We present the results in the Appendix A.2 of our updated manuscript. The evaluations show that the ResNet18 model outperformed all other models on the interference classes, prompting us to select ResNet as the backbone model.
>
> 4. We will provide evaluation results with standard deviations in the revised manuscript.

---

> > ### Comment · Reviewer_NhZE · 2025-02-21
> > **Response to Authors**
> >
> > Thank you for your responses and updated manuscript.
> >
> > Regarding my four requests:
> > R1) Thank you for the ImageNet results. It is clear that MEMO is better for ImageNet 10 task. I am grateful the authors are transparent with reporting this result - your number is still second best. A method can be an interesting contribution, even if not SOTA on all metrics for all datasets.
> > R2) Thank you
> > R3) My intention was to request experiments on a more modern architecture for all methods; however, I see that my request is not clear. The authors did a great job in showing performance of their method across a wide variety of architectures, which equally resolves my weakness. Thank you for the very thorough work.
> > R4) Could you please point out exactly where the evaluation results with STDEV are in the revised manuscript? I apologize that I could not find this with a few looks (I did not re-read the entire paper end-to-end, but I looked at results and tried to search for key words). Typically, this is done with a table that shows mean +- stdev for each result, or a plot with bars/lines/shading to show the stdev region.
> >
> > Thank you again for the hard effort. FYI, sometimes it is helpful to the reviewers for you to denote all of the changes you made in the revisions. E.g., use red or blue text to show the updates. This makes it easier for us to trace the revisions and give you all of the appropriate credit for your hard work.

---

> > > ### Author Response · Authors · 2025-02-23
> > > **Answer to Reviewer NhZE**
> > >
> > > Thank you for your feedback and additional questions.
> > >
> > > R1) We appreciate your thoughtful feedback and agree that BLCL is a valuable approach, even if it does not achieve superior performance across all metrics on the ImageNet dataset.
> > >
> > > R3) Thank you for your clarification and appreciation; we are pleased that our experiments effectively addressed your concerns.
> > >
> > > R4) The standard deviation results are presented in Table 1, Table 2, and Table 3 in Section 5 (below the BLCL results), denoted by the $\pm$ symbol. These values are relatively small, demonstrating the robustness of our method across different training runs.
> > >
> > > Additionally, we appreciate your suggestion regarding revision tracking and will ensure that future submissions clearly highlight updates to facilitate easier review.

---

### Author Response · Authors · 2025-02-17
**Official Comment on Manuscript Revision**

We would like to sincerely thank the reviewers for their time and invaluable feedback. We are dedicated to addressing the points raised and have made the necessary revisions accordingly. The updated version of the paper has now been submitted. We have provided detailed responses to the questions and concerns raised by each reviewer separately. Please let us know if any specific points have not been fully addressed.

The reviewers provided overlapping feedback, particularly regarding the motivation and evaluation of the balance between the cross-entropy (CE) and contrastive learning (CL) loss functions. The main changes are summarized as follows:

1. We have acknowledged the reviewer's feedback regarding the performance comparison and clarified that the Bayesian uncertainty weighting provides adaptability across different tasks, which is not captured by a fixed loss weighting strategy, particularly for more complex tasks.

2. We have clarified the use of the trainable hyperparameters $\sigma_1$ and $\sigma_2$ in the Bayesian weighting approach and present their update process in pseudocode in the revised manuscript.

3. We have enhanced the explanation of balancing the CE and CL loss functions, providing a detailed motivation to clarify their complementary roles and the importance of dynamic weighting throughout training in Section 3 and the Appendix A.4.

4. We have specified the adaptation of ResNet layers for new tasks in Section 3 and the Appendix A.3, detailing how the number of convolutional layers in the final block is adjusted based on class similarity.

5. Experiments involving the ImageNet dataset and standard deviations have not yet been included in the revised manuscript due to extended training times but will be incorporated during the discussion period this week.

---

### Author Response · Authors · 2025-02-21
**Official Comment on Second Manuscript Revision**

We have now provided the second update of our manuscript, incorporating all requested additional experiments. We sincerely appreciate the reviewers’ patience, as conducting additional experiments on the ImageNet dataset required several days to complete.
Below is a brief overview of the additional changes:

1. As requested by Reviewer og1a, we have included ablation experiments using only the CE or CL loss functions. For instance, in the case of the CIFAR-10 dataset (Table 2) and the CIFAR-100 dataset (Table 3), training BLCL exclusively with the CE loss results in lower performance compared to the combination of CE and CL loss. However, it still outperforms MEMO on these datasets due to the utilization of dynamic specialized blocks. Moreover, training BLCL solely with the CL loss (i.e., with a low weighting of CE) leads to significant performance degradation. These findings indicate that the combination of both loss functions is essential for achieving optimal performance.
2. As requested by Reviewer NhZE, we have added the mean and standard deviation results across multiple training runs.
3. We have conducted further experiments on the ImageNet100 dataset and included the results in Section 5, along with confusion matrix plots and t-SNE visualizations. The results demonstrate that our method significantly outperforms existing approaches and significantly outperforms MEMO on the first 5 tasks on the ImageNet dataset and enhances representation quality. While MEMO aims to enhance the CE loss, thereby achieving higher final classification accuracy, BLCL effectively learns an optimal representation by maintaining a more balanced weighting between the CE and CL loss functions. Additionally, we have included weighting plots in the Appendix, which yield conclusions consistent with those observed for other datasets.

We sincerely appreciate the reviewer’s valuable feedback and have carefully incorporated all requested changes into the manuscript. We kindly ask the reviewer to review our updated manuscript and let us know if any further modifications are needed. We believe that we have thoroughly addressed all points and that our manuscript meets all requirements while making a valuable contribution to TMLR.

---

### Author Response · Authors · 2025-03-31
**Acceptance and Camera-Ready Paper**

Dear Action Editor and Reviewers,

We sincerely appreciate your positive feedback and the acceptance of our manuscript.

In response to the requested revisions, we have made the following modifications:
1. Incorporated the discussion with Reviewer acqF in Section 5.
2. Provided a more detailed explanation of the functions Bayesian\_weighting and update_exemplars in the algorithm presented in Section 3.3.

Additionally, we have deanonymized the paper and uploaded the final manuscript along with a link to the GitLab repository.
Thank you for your time.

---

### Decision · Action_Editor_vdzs · 2025-03-16

**Recommendation:** Accept with minor revision

**Comment:**

The paper offers a meaningful contribution to the field of class-incremental learning. The proposed BLCL method is a novel approach that combines various techniques, utilizing Bayesian learning to dynamically balance cross-entropy and contrastive losses. The experimental results are generally convincing, and the authors have effectively addressed the reviewers' concerns.

The reviewers provided constructive feedback that significantly enhanced the quality of the paper. Key improvements include:
- Clarification of the Bayesian balancing mechanism: The authors provided a more detailed explanation of the Bayesian method and its advantages.
- Inclusion of ImageNet100 results: Addressing concerns about the method's applicability to higher-resolution images.
- Provision of standard deviations: Enhancing the statistical rigor of the results.
- Ablation studies: Providing insights into the contribution of individual loss components.
- Pseudocode: Improving the clarity and reproducibility of the method.

While the paper has improved significantly during the review process, two modifications should be made before final acceptance:
1. Include the discussions between Reviewer acqF and the authors in the paper, as these discussions provide valuable insights that help explain the results.
2. To ensure the content is self-contained, lines 20 and 27 in Algorithm 1 need detailed descriptions and definitions for the functions used.

The authors have shown a strong commitment to addressing feedback and enhancing their work. The reviewers' feedback, along with the authors' responses, indicate that the study is technically sound and relevant to the TMLR audience.

**Audience:**

The findings of this paper would be of interest to the TMLR audience. Class-incremental learning is a classic yet challenging problem in continual learning, and the proposed BLCL method offers an intriguing approach and analysis for continual feature representation learning.
- The method combines contrastive learning with Bayesian learning for dynamic loss balancing, which is a technique that could be of interest to researchers working in continual learning, representation learning, and Bayesian methods.
- The paper includes experiments on a real-world GNSS dataset, which increases the practical relevance and impact of the work.
- While the reviewers raised valid concerns, they generally acknowledged the potential of the method, finding the results and analysis informative.
Therefore, the paper satisfies the "interest" criterion for TMLR.

**Claims And Evidence:**

The claims made in the submission are generally well-supported by evidence. The authors conducted extensive experiments across multiple datasets, including CIFAR-10, CIFAR-100, ImageNet100, and a GNSS dataset, demonstrating the effects of their proposed BLCL method both quantitatively and qualitatively. During the review process, the reviewers raised several concerns, which the authors addressed in their responses.

- The reviewers raised concerns about the lack of results on higher-resolution images and the statistical significance of the gains. The authors addressed these concerns by including ImageNet100 results and providing standard deviations in the tables.
- Reviewers also questioned the interpretation and effectiveness of the Bayesian weighting. The authors responded with clarifications and additional analysis, including a plot of the ratio of variances to better illustrate the dynamic balancing.
- Reviewers asked for ablation studies and pseudocode, both of which were provided by the authors in the revised version.

Overall, the authors have diligently addressed the reviewers' concerns and presented sufficient evidence to support their claims.